

# Simulating measurable ecosystem carbon and nitrogen dynamics with the mechanistically-defined MEMS 2.0 model

Yao Zhang[1], Jocelyn M. Lavallee[1,2], Andy D. Robertson[3], Rebecca Even[2], Stephen M. Ogle[1,4], Keith Paustian[1,2], M. Francesca Cotrufo[1,2]

[1]Natural Resources Ecology Laboratory, Colorado State University, Fort Collins, CO 80523, USA
[2]Department of Soil and Crop Sciences Colorado State University, Fort Collins, CO 80523, USA
[3]Shell International Exploration and Production, Shell Technology Center Houston, 3333 Highway 6 South, Houston, TX 77082-3101, USA
[4]Department of Ecosystem Science and Sustainability, Colorado State University, Fort Collins, CO 80523, USA

*Correspondence to*: Yao Zhang (yao.zhang@colostate.edu)

**Abstract.** For decades, predominant soil biogeochemical models have used conceptual soil organic matter (SOM) pools and only simulated them to a shallow depth in soil. Efforts to overcome these limitations have prompted the development of new generation SOM models, including MEMS 1.0, which represents measurable biophysical SOM fractions, over the entire root

zone, and embodies recent understanding of the processes that govern SOM dynamics. Here we present the result of continued development of the MEMS model, version 2.0. MEMS 2.0 is a full ecosystem model with modules simulating plant growth with above and below-ground inputs, soil water, and temperature by layer, decomposition of plant inputs and SOM, and mineralization and immobilization of nitrogen (N). The model simulates two commonly measured SOM pools - particulate and mineral-associated organic matter (POM and MAOM), respectively. We present results of calibration and validation of

the model with several grassland sites in the U.S. MEMS 2.0 generally captured the soil carbon (C) stocks ($R^2$ of 0.89 and 0.6 for calibration and validation, respectively) and their distributions between POM and MAOM throughout the entire soil profile. The simulated soil N matches measurements but with lower accuracy ($R^2$ of 0.73 and 0.31 for calibration and validation of total N in SOM, respectively) than for soil C. Simulated soil water and temperature were compared with measurements and the accuracy is comparable to the other commonly used models. The seasonal variation in gross primary production (GPP;

$R^2$=0.83), ecosystem respiration (ER; $R^2$=0.89), net ecosystem exchange (NEE; $R^2$=0.67), and evapotranspiration (ET; $R^2$=0.71) were well captured by the model. We will further develop the model to represent forest and agricultural systems and improve it to incorporate new understanding of SOM decomposition.

## 1 Introduction

One of the biggest challenges facing humanity is the need to halt the rise in atmospheric $CO_2$ concentrations, which requires a

combined set of actions including management of terrestrial ecosystems to not only protect existing carbon (C) stocks but to also increase net sequestration to actively remove $CO_2$ from the atmosphere (Griscom et al., 2017; NASEM, 2019). Such management strategies can only be reliably identified and implemented when guided by decision support tools and ecosystem models that can accurately predict C dynamics between plants, microbes, and soils, and their responses to environmental and



management drivers using current scientific understanding (e.g., Cotrufo et al., 2015; Lehmann and Kleber, 2015; Liang et

al., 2017a; Sokol et al., 2019). While these models should ideally be verifiable using measurements of their constituent pools and fluxes, the soil components of most historical ecosystem models were built around conceptual, rather than physically-defined, pools [e.g., RothC (Coleman and Jenkinson, 1996) and CENTURY (Parton et al., 1987)]. However, recent paradigm shifts in understanding of soil organic matter (SOM) formation and persistence have led to these belowground components of ecosystem models being redesigned (e.g., Ahrens et al., 2015; Robertson et al., 2019) (a full list of current models and their

model structure comparison is in Table S1). Ensuring that the soil pools and fluxes are measurable is particularly important if these models are to be used for estimating tradable C credits or outcome-based C sequestration incentives. These models must also simulate the entire soil profile, to account for C stocks and dynamics in deep (i.e. > 30 cm) soil layers.

Ultimately, these contemporary models that represent observed mechanisms of C and nitrogen (N) dynamics will go beyond supporting management decisions, serving also as tools for scientific enquiry, enabling testing of new hypotheses and

identification of knowledge gaps. While many models currently exist and are used for these purposes [e.g., DayCent (Parton et al., 1998), DNDC (Li et al., 1992), RothC (Coleman and Jenkinson, 1996)], and new ones are being developed [e.g., MEND (Wang et al., 2013), CORPSE (Sulman et al., 2014), MIMICS (Wieder et al., 2014), COMISSION (Ahrens et al., 2015)], arguably none fully address all of the needs. As a result, terrestrial C storage remains the largest source of uncertainty in future carbon cycle projections (Ciais et al., 2014). Despite its critical role in global biogeochemical cycling, soil organic C is not

well constrained in Earth System Models (Todd-Brown et al., 2013), highlighting the need for improved simulation of plant-microbial-soil C feedbacks.

Few soil biogeochemical models have made measurable SOM pools a focal point (Abramoff et al., 2018; Fatichi et al., 2019; Robertson et al., 2019; Wang et al., 2013), despite their importance for guiding model development and judging model performance. To our knowledge, only MEMS 1.0 (Robertson et al., 2019) used measured SOM fraction data for model

calibration and verification (Table S1), while other models continue to calibrate and validate them against total soil C. Many conventional SOM models such as RothC, and DayCent do not model measurable SOM pools, and therefore attempts to validate their size have required abstraction based on measurable fractions (e.g., Zimmermann et al., 2007). Instead, these models partition total SOM into discrete pools based on turnover times, but differ in their approaches to simplify the complex mechanisms that govern SOM dynamics. Consequently, simulations of SOM pools and resulting total soil C stocks can vary

greatly between models, sometimes predicting contrasting responses to the same driving inputs and environmental change (Smith et al., 1997; Todd-Brown et al., 2014). The use of physically defined, measurable pools allows for detailed data-model comparison during parameterization and validation, with the potential to produce more accurate models that better reflect real-world processes. Many methods for separating SOM into fractions with different biogeochemical properties and turnover rates currently exist, but simple physical separations yielding two to four SOM fractions and including some form of particulate

OM (POM) and mineral-associated OM (MAOM) are widely used (Christensen, 2001; Cotrufo et al., 2019; Poeplau et al., 2018). These fractionation methods are relatively inexpensive and simple to perform, while yielding fractions with contrasting



formation and decomposition processes (Lavallee et al., 2020), making them ideal candidates for representation in biogeochemical models.

Carbon dynamics and stock distribution between POM and MAOM are linked to N (Cotrufo et al., 2019). Moving beyond C-only models to coupled C and N dynamics enables representation of mechanistic feedbacks, such as N limitation of litter decomposition (Craine et al., 2007; Knorr et al., 2005; Zhang et al., 2008) and microbial C use efficiency (CUE) (Liu et al., 2018; Sinsabaugh et al., 2016; Soares and Rousk, 2019). Additionally, it provides constraints on C and N flows according to well-known stoichiometric relationships (Buchkowski et al., 2019; Kyker-Snowman et al., 2020). Many models that include both C and N calculate N fluxes based on donor pool sizes and are constrained by the C:N ratios of receiving pools, with little or no representation of the microbial processes that control N dynamics. While this method is relatively simple and parsimonious, it fails to capture plant-microbial feedbacks that regulate N flows. For example, microbiota may alter exoenzyme production or mine SOM (Mooshammer et al., 2014) to access N to meet their needs, and plants may increase exudate production to stimulate these processes (Tian et al., 2019). Failing to represent N dynamics resulting from plant-microbial-feedbacks may lead to inaccuracies in model predictions. Emerging models such as the MIMICS-CN have begun to represent these processes in greater detail (Kyker-Snowman et al., 2020), but most ecosystem models continue to use more simplified, microbial-implicit structures to simulate N dynamics.

Physicochemical and biological properties differ markedly between subsoils (e.g., >30 cm deep) and topsoils, and there is increasing evidence that models of soil C and N storage and cycling should consider topsoils and subsoils separately. Subsoils hold more than half of the total soil C (Batjes, 2014; Harper and Tibbett, 2013), and SOM formation and stabilization processes differ from topsoils because key properties including soil texture and primary inputs to SOM, i.e., plant inputs *versus* vertical transport of dissolved organic matter (DOM), vary with soil depth (Rumpel and Kögel-Knabner, 2011). Despite their importance for C storage, sensitivity to perturbation, and the remarkable differences from topsoils, only a few recent ecosystem models explicitly represent subsoil C dynamics (e.g. Ahrens et al., 2015; Camino-Serrano et al., 2018; Fatichi et al., 2019). Most commonly, subsoil is modelled as an extension of topsoil with very limited if any validation (Braakhekke et al., 2013; Ota et al., 2013; Wieder et al., 2014), largely because of a paucity of subsoil data.

Incorporating emerging understanding of soil biogeochemical processes into models has the potential to improve model performance and increase their utility for hypothesis testing and predictions. Microbial processing of plant inputs is a key process by which SOM is formed, and mechanistically links SOM pools and plant litter quality, microbial CUE and C:N stoichiometry. For example, labile, water-soluble litter components are more likely to be processed by microbes with relatively high efficiency, forming proportionally more MAOM than structural litter components (Cotrufo et al., 2013; Haddix et al., 2016; Lavallee et al., 2018). This has been termed the *in vivo* pathway, but MAOM may also form directly from plant inputs by an *ex vivo* pathway that bypasses microbial processing (Liang et al., 2017a). The relative importance of these pathways is thought to vary greatly between the rhizosphere and the bulk soil, with *ex vivo* MAOM production playing a larger role in the bulk soil, where the density of microbial cells is lower and DOM has less chance of being intercepted prior to mineral association (Sokol et al., 2019). Inputs to SOM also differ between the rhizosphere and bulk soil, with aboveground plant





inputs only contributing appreciably to SOM in the bulk soil due to its spatial separation from roots and their exudates, which are the predominant inputs to SOM formation in the rhizosphere (Sokol and Bradford, 2019). Though these ideas have gained recognition with the scientific community and have spurred significant experimental work and hypothesis testing, to our knowledge no soil biogeochemical models yet represent all of these recent advances simultaneously.

The MEMS 1.0 model (Robertson et al., 2019) is a soil carbon model with physically defined pools that was developed in accordance with recent advances in SOM dynamics including the Microbial Efficiency-Matrix Stabilization (MEMS) hypothesis (Cotrufo et al., 2013) and interactions between litter chemistry and MAOM saturation behaviour (Castellano et al., 2015). Here we present MEMS 2.0, which builds on MEMS 1.0 to form a complete ecosystem model including N cycling, soil vertical water flows, DOM transport, plant growth, root input, and soil temperature dynamics. MEMS 2.0 represents distinct

plant inputs and microbial processes in the litter layer and rhizosphere, and DOM, POM and MAOM dynamics in the bulk soil to a user-defined depth above the bedrock. We describe the model structure, parametrization, and verification against measured ecosystem fluxes as well as soil C and N fractions along the full soil profile at multiple US grassland sites from the National Ecological Observatory Network (NEON). We focused on grasslands for this first full ecosystem version of the MEMS model because grasslands are among the largest terrestrial biomes in the world, and temperate grassland soil contains 15% of the

global soil organic C stocks (Watson et al., 2000). Grasslands have been the focus of major long-term biogeochemical research that make them an ideal "model-system" to address questions related to soil C dynamics.

## 2 Methods

### 2.1 Model description

Starting from the MEMS 1.0 version (Robertson et al., 2019), we developed a one-dimension ecosystem model, MEMS 2.0

(Fig. 1), which simulates litter layer, rhizosphere, and bulk soil C, N, water, and temperature, as well as plant growth. The main required inputs are daily weather (maximum temperature, minimum temperature, and precipitation; solar radiation is optional), soil properties, plant characteristics, and management practices. The model produces outputs on a daily time step. The soil water, soil temperature, plant and microbial nitrogen uptake, and bioturbation processes run on a sub-daily time step (hours) for higher prediction accuracy. A one-dimension soil profile is divided into continuous soil horizons in the model input

file, with user-defined depths for each horizon. While executing a model simulation, the user-defined soil horizons are further divided into thinner layers to effectively solve partial differential equations. The model simulates a surface litter layer which interacts with the first soil layer (Fig. 1). In each soil layer, the space is conceptually divided into rhizosphere and bulk soil, though there is no explicit spatial division due to the one-dimensional structure. Each simulated pool, including plant organs and soil organic pools, have both C and N components. MEMS 2.0 is coded in Java with an object-oriented structure (Fig. S1).

A full list of all model equations and the corresponding variables and parameters can be found in Tables S2, S3, and S4.



### 2.1.1 Litter decomposition

MEMS 1.0 incorporated the Litter Decomposition and Leaching (LIDEL) model (Campbell et al., 2016). In MEMS 2.0, we modified this submodel to explicitly represent the depolymerization of hydrolysable and unhydrolysable litter, and microbial uptake of DOM, turnover and contribution to litter pools. Similar to MEMS 1.0, both aboveground and belowground plant litter is divided into three pools based on its physicochemical structure (Fig. 1.). The water-soluble pool is determined as the hot-water extractable fraction of the initial litter, which is continuously replenished during litter decomposition by the depolymerization of the structural litter components (Soong et al., 2015), and is contributed by the water-soluble components of microbial biomass turnover, as described below. The litter structural component is separated into a hydrolysable pool, representing polymers, such as proteins and celluloses, and an unhydrolysable pool representing lignin, suberin, cutin, and microbial polysaccharide-lignin complexes (McKee et al., 2016). These litter fractions are commonly measured in decomposing litter or forage analyses (Rowland and Roberts, 1994; Soest et al., 1991). Both structural litter components in the litter layer and rhizosphere produce DOM through their depolymerization and are contributed by structural microbial components as microbes turn over in the litter layer and rhizosphere, respectively (Fig. 1).

Similarly to MEMS 1.0, the depolymerization and decomposition processes follow a first-order decay with rate modifiers as multipliers:

$$-\frac{dC_i}{dt} = k * m_i() * C_i \qquad (1)$$

where $C_i$ is a carbon pool in the i-th layer, $k$ is the decay rate, and $m_i()$ is a function of the multiplication of the individual modifiers for the i-th layer. For the aboveground soluble and hydrolysable pools, the modifiers are normalized functions of temperature, moisture, lignocellulose index [LCI, defined as the ratio between acid-insoluble and acid-soluble + acid-insoluble, following Soong et al., (2015)], and microbial C:N ratio (Table S2). The unhydrolysable pool does not include the LCI modifier. The belowground soluble pool contributes together with the root exudate to the rhizosphere DOM which decomposes as described above for the aboveground soluble litter pool.

Both the litter layer and the rhizosphere have a microbial biomass pool. Microbes assimilate C from the soluble and DOM pools in the litter layer and the rhizosphere, respectively. Microbial assimilation of C uses the concept of CUE which is calculated dynamically as a function of substrate C:N ratio.

$$CUE = micCN_{max}/(CN_{substrate} + CN_{CUE\_km}) \qquad (2)$$

where $micCN_{max}$ is the maximum C:N ratio of microbes, $CN_{substrate}$ is the substrate C:N ratio, and $CN_{CUE\_km}$ is a curve-adjusting parameter. The substrate C:N ratio calculation includes the organic N in the pool as well as the available mineral N. Any C taken up by microbes from the soluble and DOM pool that is not assimilated (based on CUE) is respired as $CO_2$. If the N from the substrate is more than the potential N demand of microbes, net mineralization occurs. Otherwise, there is





immobilization that consumes mineral N. The C:N ratio of microbes is dynamic as a result of CUE and the N availability from

organic and mineral sources. Microbial death also follows a first order equation and the necromass splits between soluble, hydrolysable, and unhydrolysable litter pools. The litter decomposition model was created first as a stand-alone model and was incorporated into MEMS 2.0 after verification with measured data from Soong et al., (2015).

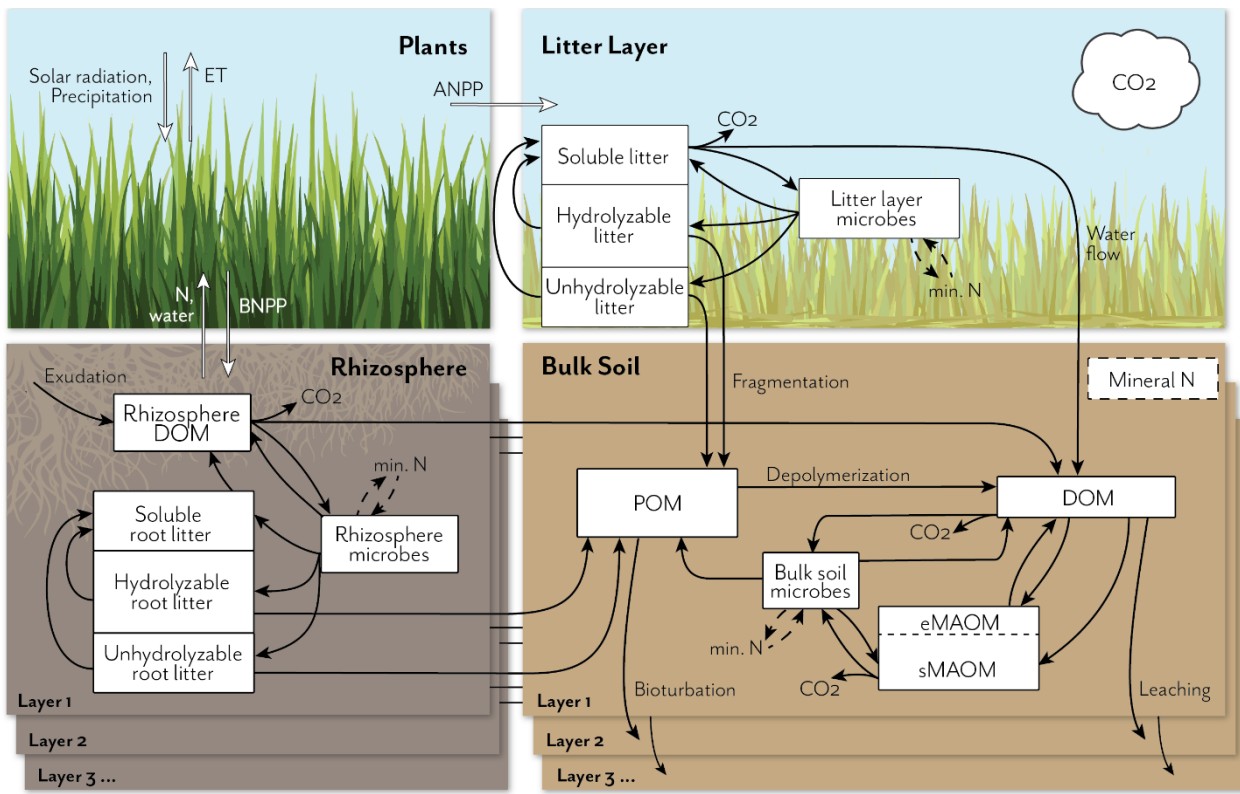

**Figure 1. Schematic representation of the MEMS 2.0 ecosystem model, showing detailed pools and fluxes for the litter and soil**
**components. The full model represents carbon (C) and nitrogen (N) fluxes among atmosphere, plants and soil, in multiple soil layers down to a user-defined depth. Inputs and recycling of N cause feedbacks to net primary productivity (NPP), which is allocated above (ANPP) or below-ground (BNPP) and at different depths, depending on vegetation and soil traits. Plant C and N litter inputs (simulated in the plant growth submodel, not shown in detail here) are allocated to three different measurable detritus pools that differ in their solubility and chemical structure. Root exudates contribute to the rhizosphere dissolved organic matter (DOM) pool.**
**These plant input pools lose mass through leaching, microbial catabolism/anabolism and fragmentation, with different rates depending on the pool C:N chemistry, temperature sensitivity, and mineral N and water demand/availability. Microbial and plant debris contribute to three physically defined and measurable soil organic matter pools, according to current understanding [e.g. the dual-pathway model of SOM formation (Cotrufo et al., 2015), in-vivo *versus* ex-vivo microbial processing (Liang et al., 2017b) and point-of-entry (Sokol et al., 2019)]: soil DOM, Particulate Organic Matter (POM) and Mineral Associated Organic Matter (MAOM)**
**The MAOM consists of exchangeable and stable component pools (eMAOM and sMAOM, respectively). Microbial pools immobilize and mineralize N, which feeds back to plant production and soil biogeochemical processes. Multiple soil layers are represented by the same belowground model structure, with DOM and mineral N moving through the soil profile and roots contributing fresh inputs at depth.**





### 2.1.2 Bulk soil organic matter dynamics

The model has five organic matter pools in the bulk soil (Fig. 1). The POM is defined either by density as $< 1.85\text{-}1.6$ g cm$^{-3}$, or by size as $> 50\text{-}60$ μm after aggregate dispersion (Lavallee et al., 2020). Inputs to the POM pool are from the fragmentation and incorporation of the structural plant and microbial litter components into the bulk soil (Cotrufo et al., 2015), from the aboveground litter layer for the top soil layer, and the rhizosphere for all soil layers. Bioturbation (soil mixing) is simulated as POM moves downward in the soil profile using an equation with the same form as the diffusion equation (Elzein and Balesdent, 1995) (Equation S12 in Table S2).

The bulk soil DOM pool receives inputs from the aboveground litter soluble pool (for the top layer only), from the rhizosphere DOM, from depolymerization of POM and from desorption of MAOM. In subsurface soil layers, DOM can also leach from layers above as an input to the bulk soil DOM pool. In the rhizosphere, a greater fraction of the DOM pool is taken up by microbes (*in vivo* pathway) *versus* being exported to bulk soil DOM without microbial processing (DOM transport from rhizosphere to bulk soil is simulated as a diffusion process controlled by soil water content and a diffusion coefficient; Equation S9). In the bulk soil, a greater proportion of the DOM pool can directly enter the MAOM pool via the *ex vivo* pathway, in accordance with the point-of-entry hypothesis (Sokol et al., 2019).

The MAOM pool is modelled as two pools: exchangeable MAOM (eMAOM) and stable MAOM (sMAOM) (Schrumpf et al., 2020). Both eMAOM and sMAOM have an upper limit of saturation that is calculated based on soil clay and silt content (Hassink, 1997; Six et al., 2002). Inputs to eMAOM are from DOM, assuming it can adsorb to mineral surfaces or existing organo-mineral associations with weak bonding (Kleber et al., 2007) that is reversible. This process is modelled using the Langmuir isotherm that assumes instantaneous equilibrium between adsorption and desorption (Mayes et al., 2012). The DOM can also associate with mineral surfaces through strong bonding, forming sMAOM. This adsorption rate is modelled as a function of water-filled pore space (WFPS), sand content, and the saturation level of sMAOM (Equation S15). The sMAOM primarily receives inputs from microbial external polymeric substances (Kleber, 2015), which are thought to be strongly protected by mineral association, but which can be slowly consumed by microbes through direct access (i.e., no DOM intermediary; Equation S15). Bulk soil microbes assimilate DOM, and their turnover contributes to POM, sMAOM, and DOM, with $CO_2$ as a byproduct based on microbial CUE (Equation S14). The depolymerization of POM, decomposition of MAOM, and decomposition of DOM follow Equation 1.

### 2.1.3 Soil temperature, water, and solutes

Soil surface temperature in MEMS 2.0 is calculated following Parton et al. (1998). It is a function of air temperature, litter biomass, plant biomass, and snow depth. The soil surface temperature serves as the upper boundary condition for the soil temperature calculation. To calculate soil temperature we adopted the method by Bittelli et al. (2015) to numerically solve the heat transport equation.



Water and solute transport are calculated simultaneously using the model described in Ross (2003). This fast and simplified method numerically solves the Richard equation for water transport and the advection–dispersion equation for solute transport (Bittelli et al., 2015). The Brooks and Corey (1964) method is used to describe the soil hydraulic properties. If the soil hydraulic

parameters are not provided by the user, the model estimates the parameters based on soil texture, bulk density, and SOM content using a pedo-transfer method (Saxton and Rawls, 2006). The litter layer is assumed to hold water based on a concept similar to field capacity and residual soil water content (SWC) (Ogée and Brunet, 2002). In MEMS 2.0, solutes simulated are DOM, ammonium, and nitrate. Mineral N can be uptaken by plant root and microbes. We modelled the competition between plants and microbes based on their demand (the amount N required to reach maximum N content) using an hourly time step

and assuming equal opportunity at each time step (Kuzyakov and Xu, 2013).

MEMS 2.0 calculates potential evapotranspiration (ET) using calculated reference ET from a grass reference surface, combined with a dynamic crop coefficient for specific plant types (Allen et al., 1998). The reference ET is estimated with the Hargreaves method that uses daily maximum and minimum temperature data as inputs (Hargreaves and Allen, 2003). The calculation of potential evaporation and transpiration is adapted from Zhang et al. (2018) and Raes et al. (2009) that uses

estimated canopy cover from leaf area index (LAI) and crop coefficients. Actual evaporation and transpiration are outputs of the soil water submodel as described above.

### 2.1.4 Plant growth

The plant growth submodel is modified from the Light INTerception and UtilLisation Version 5 (LINTUL5) crop model (Wolf, 2012) and works on a daily time step. In the MEMS 2.0 plant growth submodel, both annual and perennial herbaceous plants

can be simulated. Dry matter accumulation or net primary production (NPP) is simulated using radiation use efficiency (RUE). The model can also directly use daily NPP as an input driving variable. Estimation of plant respiration uses the method from Yin and Laar (2005).

Aboveground plant components include leaves, stems, and seeds. Belowground components are coarse roots, fine roots, and exudates. Partitioning between roots and shoots is based on species or crop variety-specific parameters defining dry matter

allocation from emergence to maturity. The partitioning of aboveground dry matter to leaves, stems, and seeds adopts the method in Zhang et al. (2018). Crop phenology is calculated based on heat accumulation and photoperiod (Soltani and Sinclair, 2012; Yin and Laar, 2005). Root distribution is modelled using the simple curve from MEMS 1.0 (Robertson et al., 2019). Rooting depth increases from plant emergence to the end of the plant growth phase as a function of phenological development. Root exudation is a species or variety-specific fixed fraction of C allocated belowground. A full list of the parameters is in

Table S4.

### 2.1.5 Fire events

Fire has a significant impact on C and N cycling in grasslands (Ojima et al., 1994; Soong and Cotrufo, 2015), and we included a simple fire module in MEMS 2.0. We acknowledge that fire deserves a more detailed representation, including the production



of pyrogenic organic matter (PyOM) and its cycling in soil (Bird et al., 2015; Knicker, 2011). However, a more detailed

representation of fire impacts on C was beyond the scope of this model version, and we intend to develop a PyOM module in

a future version.  In MEMS 2.0, natural and prescribed fire events can be scheduled in the management schedule input file of

the model. We used an approach modified from the DayCent model (Hartman et al., 2020; Ojima et al., 1994). A fire event

removes aboveground live and dead biomass and surface litter, according to user-defined percentages of each pool (default

values of 60%, 80%, and 80% removal for aboveground live biomass, standing dead, and litter, respectively). Pyrogenic C is

returned to the soil surface and added to the unhydrolysable pool of the aboveground litter layer. A user-defined fraction of

the N in burned plant biomass can be also returned to soil surface (Hobbs et al., 1991).

## 2.2 Observation datasets used for model calibration and validation

We calibrated the litter decomposition model using a dataset from a laboratory incubation of a range of litter types (Soong et

al., 2015). Leaf litters covering a wide range of C:N ratios and LCI values were collected and incubated for one year. During

this time, DOM was leached and measured periodically and the total dry matter remaining was recorded twice on day 95 and

365.

Soil samples from the NEON sites (Hinckley et al., 2016) were used to calibrate and validate the model for this study (Fig. 2,

Table 1). The NEON Megapits are single, temporary soil pits used to characterize soils and collect initial soil samples at each

site. Archived air-dried Megapit soils were obtained from NEON and fractionated to quantify C and N in DOM, POM, and

MAOM. The fractionation scheme produced four fractions after aggregate dispersion: DOM (water soluble, < 20 µm), light

POM (< 1.85 g cm$^{-3}$), a heavy coarse fraction (> 53 µm and > 1.85 g cm$^{-3}$), and heavy fine fraction (< 53 µm and > 1.85 g cm$^{-3}$). Since MEMS 2.0 represents only three bulk soil SOM fractions, we added the two heavy fractions into the modelled MAOM

fraction. This approach appeared more reasonable for these soils because of the relatively low C:N ratio of the heavy coarse

fraction (first quantile 7.9 and third quantile 15.0), which was more similar to the MAOM (first quantile 7.3 and third quantile

12.7) than to the POM (first quantile 18.6 and third quantile 42.44). We still lack a complete understanding of the heavy coarse

SOM fraction, and mechanistic research is required to clarify its role and function in soil (Lavallee et al., 2020). The soil

samples were air-dried and sieved to 2mm by NEON according to their soil archiving protocol prior to being shipped to

Colorado State University. Upon receipt, all soil samples were heated at 116 °C for 18 hours. Subsamples (5.5-6.0 g) were

weighed into clean 50 mL conical centrifuge tubes, and placed in a 60 °C oven overnight to minimize any excess moisture.

Each tube received 35 mL of deionized water, was placed on a reciprocal shaker for 15 minutes, and then centrifuged at 1874

g for 15 minutes. The water and DOM were decanted over a 20 µm nylon filter and frozen at -20°C until analysis. Light

particulate material caught on the filter was set aside as light POM and oven dried at 60 °C. Each tube then received 25 mL of

sodium polytungtate (SPT) at 1.85 g cm$^{-3}$ along with 12 glass beads and was vortexed to resuspend the pellet. Tubes were

placed on a reciprocal shaker on low speed for 18 hours to disrupt aggregates. After dispersion, the tubes were centrifuged at

1874 g for 30 minutes, and the light POM in suspension was aspirated onto a 20 µm filter using a vacuum filtration system

and oven dried at 60 °C. The pellet was then rinsed multiple times with DI water to remove any residual SPT. For each rinse,

each tube received deionized water, was shaken to disrupt the pellet, and centrifuged at 1874 g for 20 minutes or until the supernatant was clear. Finally, the remaining heavy fraction was sieved at 53 µm to separate the course (> 53 µm) from the fine (< 53 µm) heavy fraction and both fractions were oven dried at 60 °C. Dried soil fractions were finely ground to ensure
homogenization before being analysed for C and N content on an elemental analyser (Costech ECS 4010; Costech Analytical Technologies; Valencia, CA, USA).

**Figure 2. Locations of the sites used for model calibration and validation. The sites are part of the National Ecological Observatory**
**Network (NEON), Soil Climate Analysis Network (SCAN), or Ameriflux Network.**

**Table 1. Characteristics of the six sites used to calibrate and validate MEMS 2.0. Soil properties (average of top 20 cm) are from the Megapit soil samples of the National Ecological Observatory Network (NEON). Site ID are from the NEON database.**



| Site ID | Grass Type | MAT[1] (°C) | MAP[2] (mm) | NPP[3] (g C m$^{-2}$) | Soil Family | Sand (%) | Clay (%) | pH (CaCl$_2$) | Historical Fire Interval[4] (Years) |
|---|---|---|---|---|---|---|---|---|---|
| CPER | Shortgrass | 8 | 370 | 216 | Fine, loamy, mixed, superactive, mesic. Aridic Argiustolls. | 71.5 | 11.9 | 7.3 | 13 |
| DCFS | Mixed | 5 | 490 | 501 | Fine, loamy, mixed, superactive, frigid. Typic Haplustolls. | 31.5 | 34.5 | 6.2 | 13 |
| KONZ | Tallgrass | 12 | 860 | 381 | Fine, smectitic, mesic. Pachic Udertic Argiustolls. | 5.3 | 36.5 | 5.9 | 7 |
| NOGP | Mixed | 5 | 400 | 394 | Fine, loamy, mixed, superactive, frigid. Typic Argiustolls. | 17.5 | 25.1 | 6.1 | 13 |
| OAES | Mixed | 15 | 670 | 260 | Loamy, mixed, active, thermic. Lithic Haplustepts. | 12.7 | 17.0 | 7.6 | 4 |
| WOOD | Mixed | 5 | 490 | 450 | Coarse, loamy, over-sandy or sandy, skeletal, mixed, superactive, frigid. Typic Haplustolls. | 57.3 | 18.1 | 6.8 | 13 |

[1] Mean annual temperature.
[2] Mean annual precipitation.
[3] Mean annual MODIS NPP (Running et al., 2015).
[4] Historical fire interval data was from Guyette et al. (2012). Frequency of prescribed fire in the experimental period was different and obtained from the experimental records of individual sites.

Five year SWC and soil temperature data at four depths from the Soil Climate Analysis Network (SCAN) (Schaefer et al., 2007) were used to verify the model representation of moisture and temperature dynamics. For this evaluation, we used four SCAN sites in the Great Plains representing wet, dry, hot, and cold climate with various soil textures (Fig. 2). In addition, the model was tested with multiple years of eddy covariance (EC) flux data (2007-2010), including net ecosystem exchange (NEE), gross primary production (GPP), ecosystem respiration (ER), and actual ET from an EC tower at the Konza Prairie (Ameriflux ID US-Kon; Fig. 2) (Brunsell et al., 2014). Although NEON collects Eddy Covariance flux data (NEON, 2020), the data were not used for model testing in this study because available data are incomplete and from short monitoring periods.

## 2.3 Model set up, calibration and validation

The six NEON grassland sites (Fig. 2) were used for testing model performance of C and N dynamics. These grassland sites were assumed to be at a steady state in terms of soil C and N stocks, and a long–term simulation was conducted to reach steady state for comparing the model output to the measurement data (Robertson et al., 2019). In our testing, three hundred years was determined to be the minimum time needed for the system to reach steady state. The model input data were collected from several different sources (Table S5). Our measured MAOM C data from all NEON sites were used for deriving the parameters for the saturation function (Hassink, 1997; Robertson et al., 2019) by fitting a boundary line to the data (Feng et al., 2013).





The daily NPP derived from a MODIS product was used as input data in these simulations (Table S5) (Running et al., 2015).

Other plant growth-related parameters were set based on measurements and literature (Table S5). Although some of the NEON sites are under livestock grazing, this version of model does not simulate grazing and its effect on plant growth and organic matter decomposition (in future development plan). The input data of MODIS NPP could reflect some the effect of grazing on plant production.

The SCAN sites were set up using their site-specific weather and soil data (Table S6). The specific soil texture of the EC site

at Konza was not available. As the nearby NEON KONZ site soil has the same soil texture class (silty clay loam) so soil parameters from NEON KONZ were used for the EC site. The plant-related parameters were based on the nearest NEON grassland site. The plant production parameters for modelling the EC site was adjusted using the GPP data of the first measurement year (2007) to reflect the productivity at this specific location (NPP was simulated by the model).

To test the litter decomposition module using the lab experiment of litter decomposition for five types of plants in Soong et al.

(2015), we created a stand-alone litter decomposition model written in R. An automated calibrations were conducted to calibrate the stand-alone litter decomposition model. In a second calibration, the MEMS 2.0 model was calibrated with NEON grassland site data. The calibration method used was a Markov Chain-Monte Carlo (MCMC) Bayesian approach, specifically the DifferRential Evolution Adaptive Metropolis (DREAM) (Vrugt and Ter Braak, 2011), using the DREAM package in R (Vrugt, 2016).

Prior to the calibration, a global sensitivity analysis on KONZ and CPER sites (wet and dry sites) was first conducted to select the most sensitive parameters used in calibration (Zhang et al., 2020a) based on the Sobol-Martizez method (Baudin et al., 2016) in the R package "sensitivity" (Iooss et al., 2020). The total sensitivity indices (Zhang et al., 2020a) account for interactions between parameters, but only SOM related parameters were investigated. The parameter ranges were estimated based on the values reported in the literature and/or with estimates from manual calibration (Table 2). The response variables

used in the sensitivity analysis were MAOM C and POM C in top 30 cm of the soil profile.

The six NEON sites were divided into calibration (KONZ, CPER, WOOD, and DCFS) and validation (NOGP and OAES) datasets. Our analysis follows the ecosystem model calibration and uncertainty analysis using DREAM described in Zhang et al. (2020a). The top 15 most sensitive parameters were selected for calibration for the NEON grassland sites (Table 2). The measured MAOM C and POM C in all horizons within the top 1 m of the calibration sites were used in the objective function.

Although NPP can be an input data in the model, it may still be modified by the model based on soil N. The plant growth submodel calculates plant N demand each day based on NPP, and when soil N cannot meet the plant requirement to maintain the minimum C:N ratio in biomass, the actual NPP is reduced accordingly. To prevent the reduction in NPP below the yearly total NPP, this was added in the objective function of the calibration.

All of the analyses were done in R (V3.5.1, R Core Team, 2017). Model results were evaluated using the coefficient of

determination ($R^2$), bias, and Root Mean Square Error (RMSE), which are commonly used in modelling studies (Zhang et al., 2020b).





**Table 2. Parameter names, definitions, ranges for sensitivity analysis, and calibrated optimal values. Listed in alphabetical order by parameter name. Parameters in bold were the top 15 most sensitive parameters. The list of acronyms is CUE (carbon use efficiency), DOM (dissolved organic matter), LCI (lignocellulose index), MAOM (mineral associated organic matter), eMAOM (exchangeable MAOM), sMAOM (stable MAOM), and POM (particulate organic matter).**

| Parameter Name | Definition | Unit | Optimal Value (range) |
|---|---|---|---|
| **CN_Microbe_max** | Maximum C:N of microbial biomass | g C g$^{-1}$ N | 11.3 (10, 14) |
| **CN_Microbe_min** | Minimum C:N of microbial biomass | g C g$^{-1}$ N | 6.7 (4, 8) |
| **Coeff_LitterCNOnCUE** | Coefficent used to calculate CUE as a function of substrate C:N (Equation S18) | g C g$^{-1}$ N | 5.1 (5, 15) |
| **Coeff_MoistureEffOnDecomp_1** | Coefficent used for the moisture effect on decomposition for all pools (Equation S25) | - | 95.4 (10, 150) |
| Coeff_MoistureEffOnDecomp_2 | Coefficent used for the moisture effect on decomposition for all pools (Equation S25) | - | 11.3 (9, 13) |
| **Coeff_Sorp_K** | Scaling coefficient used to esimate the binding affinity for the sorption of eMAOM pool (Equation S21) | - | 4.5 (0.01, 5) |
| **Coeff_TemperatureEffOnDecomp_1** | Coefficent used for the temperature effect on decomposition for all pools (Equation S24) | - | 18.4 (15, 30) |
| **Coeff_TemperatureEffOnDecomp_2** | Coefficent used for the temperature effect on decomposition for all pools (Equation S24) | - | 0.21 (0.2, 0.4) |
| Conductivity_bioturbation | Conductivity used for estimating bioturbation (Equation S12) | cm$^2$ day$^{-1}$ | 0.15 (0.001, 1) |
| **CUE_max** | Maximum CUE of microbes | - | 0.46 (0.45, 0.6) |
| Eff_LCIOnDecay_min | Minimum effect on litter decompostion corresponding to LCI_min | - | 0.35 (0.1, 0.5) |
| **Frac_MAOMExchangeable** | Fraction of the whole MAOM pool that is eMAOM at saturation | - | 0.078 (0.01, 0.2) |
| **Frac_MicrobeToHydrol** | Fraction of the microbial necromass allocated to the hydrolysable litter pool | - | 0.13 (0.1, 0.3) |
| Frac_MicrobeToPOM | Fraction of the microbial necromass allocated to the POM pool | - | 0.052 (0.01, 0.1) |
| Frac_MicrobeToSoluble | Fraction of the microbial necromass allocated to the soluble litter pool | - | 0.65 (0.5, 0.7) |
| **k_DOMDecay** | Maximum decay rate of DOM at optimal temperature and moisture | day$^{-1}$ | 0.95 (0.1, 1) |
| **k_DOMSorp** | Maximum sorption rate of bulk soil DOM at optimal temperature and moisture | day$^{-1}$ | 0.013 (0.01, 0.5) |
| k_HydrolDecay | Maximum decay rate of hydrolysable litter at optimal temperature and moisture | day$^{-1}$ | 0.014 (0.01, 0.05) |
| **k_MAOMDecay** | Maximum decay rate of sMAOM at optimal temperature and moisture | day$^{-1}$ | 0.00034 (0.0001, 0.005) |
| k_MicrobeDeath | Microbial death rate | day$^{-1}$ | 0.57 (0.1, 0.8) |
| **k_POMDepolymer** | Maximum depolymerization rate of POM at optimal temperature and moisture | day$^{-1}$ | 0.0033 (0.001, 0.01) |
| k_SolubleDecay | Maximum decay rate of soluble litter at optimal temperature and moisture | day$^{-1}$ | 0.37 (0.01, 1) |
| k_SolubleLeach_max | Maximum leaching rate of soluble litter to DOM pool with water | day$^{-1}$ | 0.20 (0.1, 0.5) |
| **k_StructToPOM** | Maximum litter fragmentation rate to produce POM | day$^{-1}$ | 0.11 (0.01, 0.2) |
| k_UnhydrolDecay | Maximum decay rate of unhydrolysable litter at optimal temperature and moisture | day$^{-1}$ | 0.007 (0.001, 0.02) |
| LCI_max | Maximum LCI used in the calculation of LCI effect on litter decompostion | - | 0.25 (0.2, 0.4) |





| LCI_min | Minimum LCI used in the calculation of LCI effect on litter decomposition | - | 0.064 (0, 0.2) |
|---|---|---|---|

## 3. Results

### 3.1 Evaluating the stand-alone litter decomposition model.

The litter decomposition experiment (Soong et al., 2015) showed that plant litter with a high C:N ratio and high LCI value decomposes more slowly. The stand-alone litter decomposition model represented this effect of litter chemical composition on C loss rates. The predicted litter layer C stock and cumulative dissolved organic carbon (DOC) leached during the year matched the measured values accurately in the calibration (Fig. 3). The $R^2$, bias, and RMSE were 0.96, 2.93 g m$^{-2}$, and 65.65 g m$^{-2}$, respectively for the total litter C stocks across all pools; and 0.9, -1.17 g m$^{-2}$, and 18.91 g m$^{-2}$, respectively for cumulative

DOC leached.

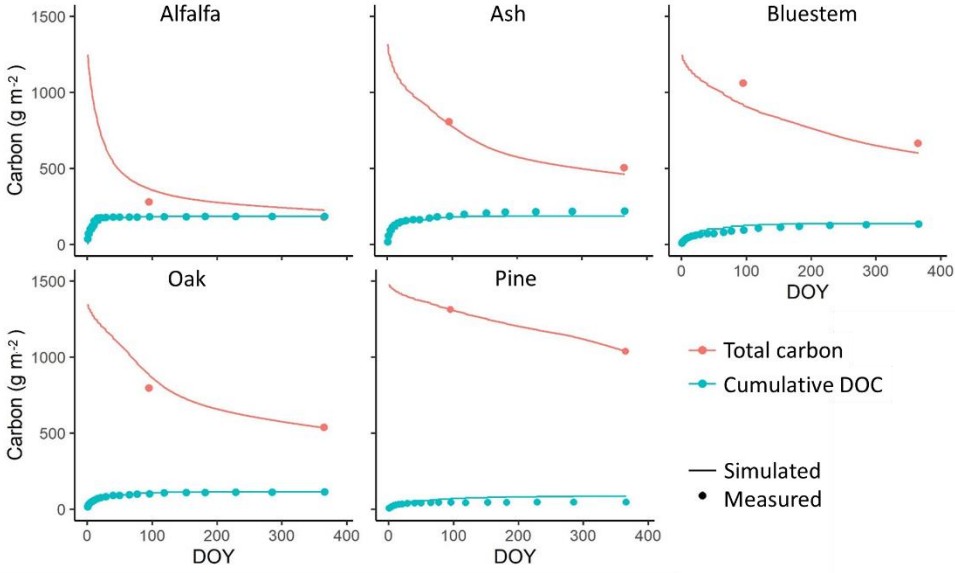

**Figure 3. Comparison between simulated and measured total litter layer carbon stocks and cumulative leached dissolved organic carbon (DOC) for a variety of litter types over one year from a one-time litter addition event using the stand-alone litter**
**decomposition model in calibration. Measured data are from Soong et al. (2015).**



**3.2 Evaluating soil organic matter predictions from MEMS 2.0**

The six NEON grassland sites cover diverse climates and soil textures (Fig.2 and Table 1), along with a wide range of soil C
concentration and fractional distributions between POM and MAOM. The highest soil C content was found at the DCFS site,
likely a result of relatively high NPP (Table1) and relatively low decomposition rates (cold and relatively dry climate). The
WOOD site is only 11 km away from the DCFS site, but the soil texture was different between the two sites and the WOOD
site was not grazed by livestock while the DCFS was grazed. We expect the large difference in soil texture (Table 1), may
explain the difference in fractional distribution in the topsoil layers at these two sites, with WOOD having a sandier texture
and a larger proportion of total C and N in POM. The CPER site is located in a semi-arid climate with a soil texture of more
than 70% sand, though it still has a relatively high proportion of total C and N in the MAOM fraction. The CPER site has the
lowest soil C content as a result of the low plant production and relatively high decomposition rates. KONZ, OAES, and NOGP
all have similar C contents and approximately equal proportions of POM and MAOM, though NOGP has slightly more C in
POM in the top soil layer. At all sites, total soil C and N concentrations decreased, and the proportion of soil C and N in
MAOM increased with depth (Fig. 4).
The calibrated model captured the distribution of C in MAOM and POM along the soil profile well for the four calibration
sites (Fig. 4; Table 3). The model also captured the distribution of the soil C for the two validation sites well, but with slightly
lower accuracy compared with the calibration data set (Table 3). The largest difference was found for the first soil horizon of
the NOGP site, where the model overestimated MAOM C by 68.7%. The model under-predicted C in the second and third
horizon of the OAES site.
Our predictions of the soil N pools were slightly less accurate compared to that of the soil C pools (Fig. 5 and Table 3). Both
the field measurements and model results showed that most (> 50% on average, especially in deeper soil layers) of the C and
N in grassland soils are stored in the MAOM fraction, which is more resistant to decomposition. As expected, the relative
contribution of MAOM increases with depth, as structural aboveground plant inputs only contribute to POM in the top layer
(Fig. 1), and structural root inputs to POM decrease with depth.

**Table 3. Statistics for the simulations of carbon and nitrogen concentrations (kg m⁻³) by layers in total, mineral associated (MAOM), and particulate organic matter (POM) pools across the NEON grassland sites used for calibration and validation of the MEMS 2.0 model.**

|  | Carbon | | | Nitrogen | | |
|---|---|---|---|---|---|---|
|  | MAOM | POM | Total | MAOM | POM | Total |
| Calibration | | | | | | |
| R² | 0.78 | 0.87 | 0.89 | 0.67 | 0.81 | 0.73 |
| bias | -0.24 | -0.62 | -1.71 | 0.12 | -0.14 | -0.02 |
| RMSE | 3.64 | 2.14 | 4.37 | 0.62 | 0.29 | 0.6 |
| Validation | | | | | | |
| R² | 0.48 | 0.59 | 0.6 | 0.16 | 0.84 | 0.31 |
| bias | 1.81 | -1.09 | 0.01 | 0.33 | -0.1 | 0.23 |
| RMSE | 4.75 | 2.82 | 6.17 | 0.86 | 0.13 | 0.84 |





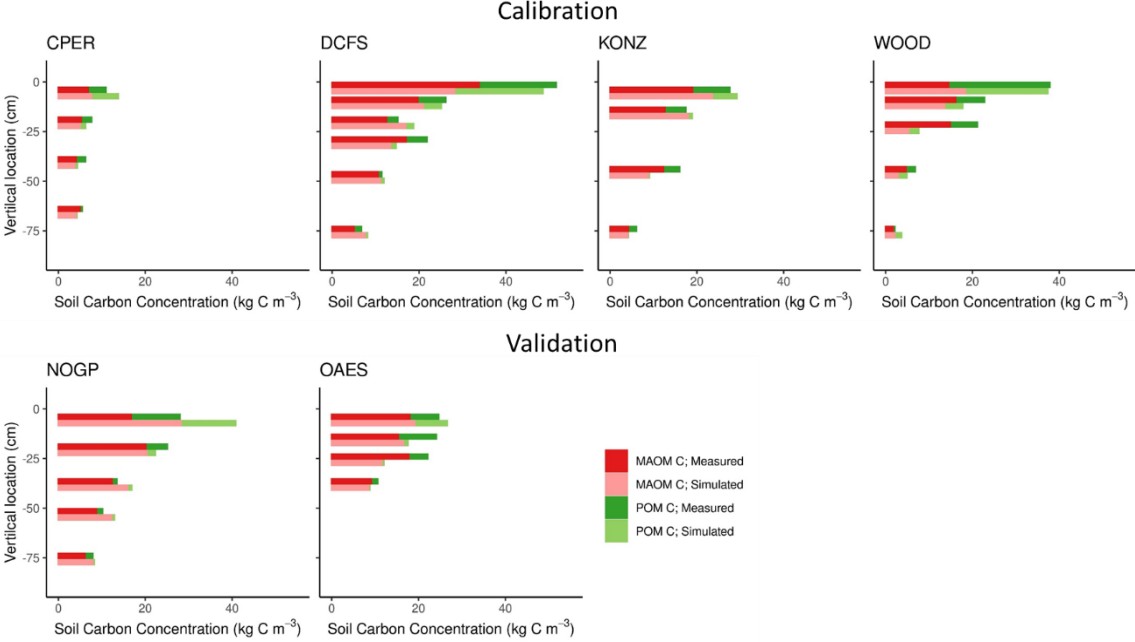


**Figure 4. Simulated and measured soil carbon (kg C m⁻³) in mineral-associated (MAOM) and particulate organic matter (POM) pools by diagnostic soil horizon for calibration and validation sites from the NEON network. Sites are described in Table 1.**

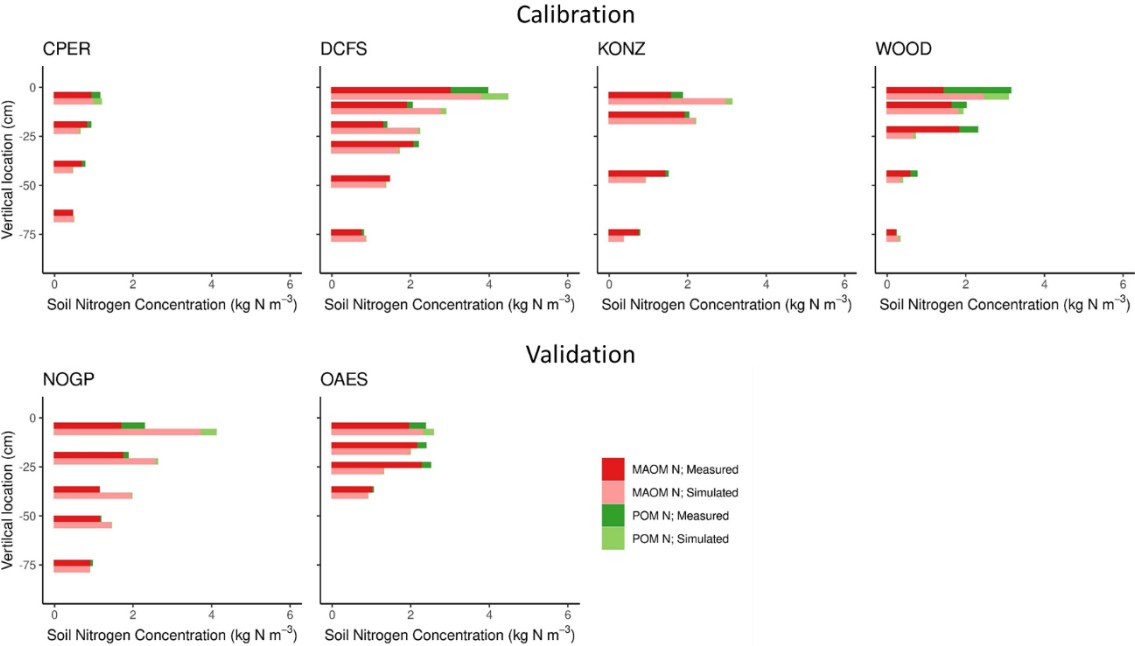

**Figure 5. Simulated and measured soil nitrogen (kg N m⁻³) in mineral-associated (MAOM) and particulate organic matter (POM) pools by diagnostic soil horizon for calibration and validation sites from the NEON network. Sites are described in Table 1.**



### 3.3 Parameter sensitivity for soil organic matter module in MEMS

A total of 27 parameters in the soil organic matter module were tested in the sensitivity analysis. The model sensitivity analysis
showed that C in MAOM at steady state was sensitive to different parameters than was POM, but the parameter rankings were
similar between sites for the same pool of C (Fig. 6). The decay rate of sMAOM (k_MAOMDecay) had the largest effect on
MAOM C, as expected. Similarly, the POM depolymerization parameter (k_POMDepolymer) had the largest effect on POM
C. The second most sensitive parameter for both MAOM and POM was the coefficient used to define the temperature effect
curve on decomposition (Coeff_TemperatureEffOnDecomp_1). Decomposition is well known to be largely affected by
temperature (Conant et al., 2011). The MAOM C is also sensitive to the fraction of total sorption capacity that is exchangeable
(Frac_MAOMExchangeable) and the sorption rate of DOM to sMAOM (k_DOMSorp), while all other parameters had
relatively low sensitivity index values. All the other parameters tested had relatively low impact on POM C. Almost all the
litter decomposition related parameters have low sensitivity indices; these parameters may substantially affect C in MAOM
and POM within a short time after litter is produced but the effect is low when the system reaches steady state. The parameters
defining the soil water effect on decomposition appeared to have a higher impact on both POM and MAOM at the dry site
CPER than at the wet site KONZ.





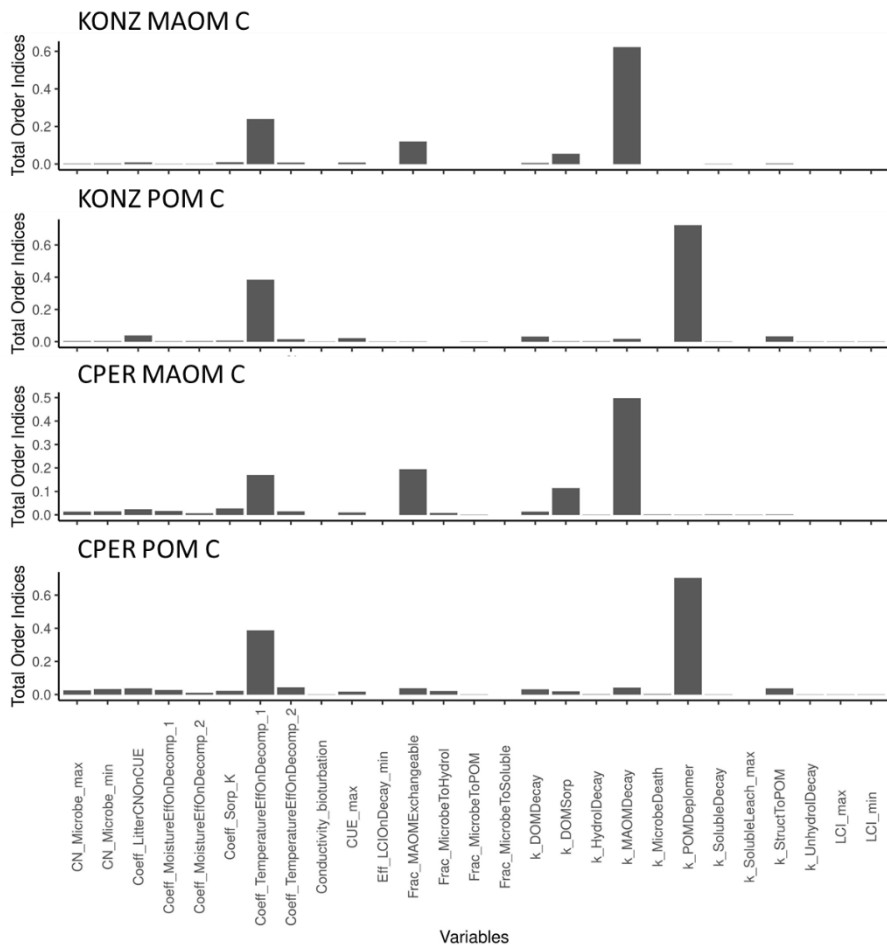

**Figure 6. Total sensitivity indices from the global sensitivity analysis for steady state C in mineral associated (MAOM) and particulate (POM) organic matter for the 0-30 cm depth at two sites. Site characteristics are presented in Table 1. Definitions of parameters are presented in Table 2.**

**3.4 Evaluating the predictions of soil water, soil temperature, and ecosystem fluxes.**

Soil temperature and moisture are the two major abiotic environmental factors controlling decomposition. Overall, the model prediction for soil temperature is relatively accurate (Table 4 and Fig. S2). The Nunn site in Colorado had lower accuracy than the other sites. Regarding soil moisture, the overall accuracy was lower than that of soil temperature. The accuracy of model predictions on SWC decreases down the soil profile (Table 4 and Fig. S2). Change in SWC further from the surface (100 cm) were small as most rainfall events at these sites did not reach deeper soil depths.





**Table 4. The statistics of simulated soil temperature and water content of four SCAN sites for model testing. Daily values in the growing season (between day of year 100 and 300) of five years (2014 – 2018) were used. Winter period is excluded because the moisture sensor measures liquid water and the model predicts total water.**

| Depth (cm) | Fort Assiniboine, MT | | | Rogers Farm, NE | | | Nunn, CO | | | Bushland, TX | | |
|---|---|---|---|---|---|---|---|---|---|---|---|---|
| | $R^2$ | RMSE | bias | $R^2$ | RMSE | bias | $R^2$ | RMSE | BIAS | $R^2$ | RMSE | bias |
| Soil Temperature (°C) | | | | | | | | | | | | |
| 10 | 0.86 | 2.67 | -1.19 | 0.67 | 3.25 | -0.14 | 0.77 | 3.75 | 0.12 | 0.82 | 2.65 | -1.39 |
| 20 | 0.90 | 2.08 | -1.12 | 0.63 | 3.04 | 0.25 | 0.74 | 3.55 | -0.31 | 0.86 | 2.19 | -1.33 |
| 50 | 0.89 | 1.85 | -1.12 | 0.43 | 3.91 | 1.80 | 0.69 | 3.42 | -0.93 | 0.84 | 1.79 | -0.97 |
| 100 | 0.87 | 2.23 | -1.70 | 0.65 | 2.57 | -0.24 | 0.54 | 5.65 | -4.77 | 0.82 | 1.85 | -1.22 |
| Soil Water Content (m m$^{-1}$) | | | | | | | | | | | | |
| 10 | 0.48 | 0.08 | -0.06 | 0.53 | 0.07 | -0.02 | 0.49 | 0.05 | 0.03 | 0.69 | 0.04 | 0.01 |
| 20 | 0.40 | 0.06 | -0.05 | 0.54 | 0.07 | -0.04 | 0.42 | 0.05 | 0.04 | 0.74 | 0.06 | -0.04 |
| 50 | 0.34 | 0.11 | -0.10 | 0.23 | 0.13 | -0.12 | 0.16 | 0.07 | 0.05 | 0.59 | 0.11 | 0.01 |
| 100 | 0.01 | 0.11 | -0.11 | 0.25 | 0.17 | -0.17 | 0.07 | 0.06 | 0.05 | 0.31 | 0.1 | -0.07 |


The model predicted measured weekly cumulative GPP with relatively high accuracy ($R^2$=0.83, bias = 2.98 g C m$^{-2}$, and RMSE= 15.46 g C m$^{-2}$). However, it slightly over predicted the peaks of GPP in summer (Fig. 7). The model also captured the changes of ER with $R^2$=0.89, bias = 6.43 g C m$^{-2}$, and RMSE= 9.02 g C m$^{-2}$. The difference between GPP and ER is NEE. As GPP was over predicted and ER was under-predicted, the NEE prediction was biased in summer periods ($R^2$=0.67, bias = 3.45

g C m$^{-2}$, and RMSE= 12.22 g C m$^{-2}$). The modeled actual ET matched the measurements with $R^2$=0.71, bias = 0.07 cm, and RMSE= 0.73 cm. The measurements from the EC tower at the Konza Prairie used in this study showed lower GPP, ER, and ET in 2009 compared with the other two years (Fig. 7). However, the model predictions did not show the same pattern. Due to the biases in predicted GPP and ER, the model underestimated NEE in the summer growing seasons of the three years.

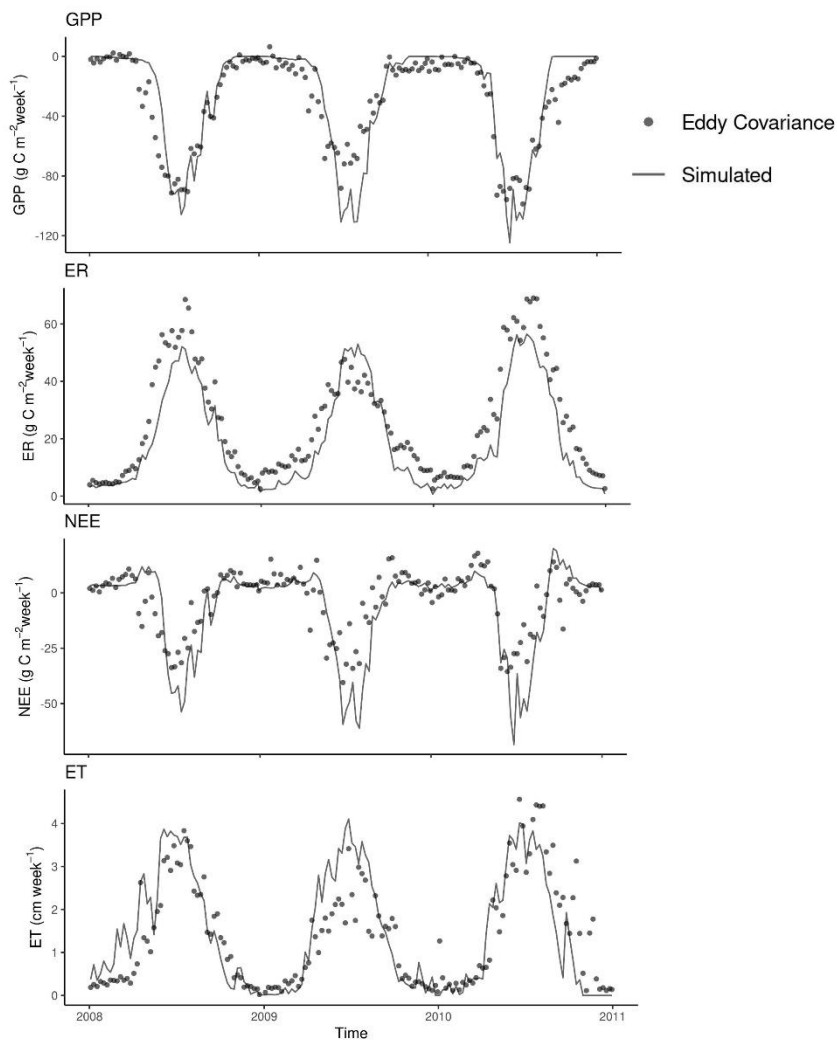


**Figure 7. Comparison between weekly gross primary production (GPP), Ecosystem respiration (ER), net ecosystem exchange (NEE) and evapotranspiration (ET) from Eddy Covariance measurements at the KONZ prairie (Table 1) and simulated (validation data only). Data in year 2007 were used for GPP calibration.**

## 4. Discussion

We built the MEMS 2.0 ecosystem model to represent state-of-the-art understanding of plant-microbial-soil C and N dynamics, using biophysically defined and measurable pools and fluxes. Our goal was to have a tool that can be improved over time to



represent different ecosystems and drivers (e.g., fire, fauna, and management) as new understanding emerges, and to be useful for scientific research inquiry as well as decision support.

MEMS 2.0 generally captured total soil organic C stocks and their distributions between POM and MAOM throughout the soil

profile (Fig. 4, Table 3). Regarding the top soil, modelled MAOM and POM C were consistent with the measurements for the calibration sites and the OAES site in validation. However, the model overestimated MAOM C at the NOGP site. Compared with the other two sites in North Dakota (DCFS and WOOD), the NOGP site has much lower measured C stocks (Fig. 4). While the annual average temperature is the same, the precipitation is 20% lower compared with the other two sites. Lower precipitation results in lower NPP and litter input to soil, likely causing the lower C stocks at the NOGP site. The NOGP site

also has lower sand content relative to the other sites, which results in a higher predicted MAOM saturation limit according to the linear equation used in the model (Equation S23) (Hassink, 1997; Six et al., 2002). The high MAOM saturation limit may explain the over-prediction of MAOM C. Although sand content (or the sum of clay and silt content) has been used for estimating MAOM saturation deficit in many studies, other studies have shown that soil texture alone may not be the best indicator (Beare et al., 2014; Curtin, 2002; Rasmussen et al., 2018).

The simulated C distribution and C fractions in MAOM and POM in deeper soil layers matched the measurements relatively well. Most traditional and new generation soil C models do not have the capacity to predict deep (i.e. > 30 cm) C dynamics. With advancements in understanding of C and N in deep soils built into MEMS, we were able to represent the distribution of C of these grassland sites across the range in climates and soil properties. The largest discrepancy between predicted and measured MAOM C in deeper layers is at 25 cm (third soil layer in Fig. 4) at the WOOD site. The reason might be again

related to the simulated MAOM saturation. The sand contents of the top two layers of this site are 55 and 59 %, but it increases to 74 % in the third layer. The low simulated MAOM saturation limit in this layer may have constrained the accumulation of C in the MAOM pool in our simulation. Overall, the model tended to underestimate POM C in the deeper layers across all sites, and there are several possible reasons for this. We use a simple representation of bioturbation by soil fauna that moves POM downward in the soil (Equation S12). We used a constant coefficient for all sites without considering differences in

climate, soil properties, and fauna community composition. It is possible that the maximum POM decomposition rate in deep soil is reduced compared with the top soil (Gill et al., 1999), but because these dynamics are not well understood, we chose to use the same maximum specific decomposition rate for all soil layers.

In general, inaccurate model predictions could also be a result of the uncertainty in model input data such as NPP, fire frequency, biological N fixation, and historical weather conditions. For example, fire could consume a large fraction of

aboveground biomass and we found that soil C was very sensitive to the amount biomass added to soil. However, there were not enough historical records to accurately estimate historical fire frequency at the sites (Guyette et al., 2012). The NPP data were derived from a MODIS product which has its own errors (Turner et al., 2006) and those errors can propagate into the errors in our simulation.

In terms of model performance, our model has achieved similar or higher accuracy compared with other models. The widely-

used Century model was validated using grassland data from the Great Plains (Parton et al., 1987) and our model achieved



similar accuracy. The SOMic model (Woolf and Lehmann, 2019) was calibrated and validated using total soil C data from three long-term agricultural experimental sites and matched measurements of top soil horizon very well ($R^2 = 0.92$), however, calibration and validation datasets used the same sites but different treatments. Total soil C values modelled by the BAMS1 model (Dwivedi et al., 2017) were compared to measured data across serval sites at different depth. The model represented the

vertical distribution of total soil C well with $R^2$ between 0.75 and 0.97 for different sites. To our knowledge, there are no other models that were calibrated and validated using measured soil C fractions from multiple sites, expect for MEMS 1.0 (Robertson et al., 2019), although measurable pools are present in many new generation models (Table S1).

Nitrogen is also an important component in SOM dynamics and affects many processes in the soil. MEMS 2.0 predicted soil N stocks reasonably but with lower overall accuracy than soil C (Fig.4; Table 3), possibly because N cycling is more complex

in soil. One of the possible causes is the uncertainty in our input data of the atmospheric N deposition and biological N fixation which are the two main external inputs of N to these grassland sites. Another is that N consumed and excreted by grazing is not accounted for by our simulations. Compared with the Century model, two studies (Parton et al., 1987, 1993) showed that the Century model simulated soil N in top soil with higher overall accuracy than ours in grasslands. However, as it is more challenging to match predictions N in SOM fractions at different soil depths, the accuracy of MEMS 2.0 on N is acceptable.

Regarding the new generation models, they either do not model N or have not been validated with measured soil N (Table S1). The method used for soil surface temperature estimation was adopted from the empirical method used in the DayCent model which has been tested in various systems (Parton et al., 1998; Zhang et al., 2013). Our validation also showed the predicted soil temperature at 10 cm matched the measurements. The model tended to underestimate soil temperature at 10 cm at Nunn, Colorado in 2014 (Fig. S2), though the bias was much smaller in other years. This may be due to an underestimation of the

amount of surface litter and live biomass in that year as the litter and biomass quantities are the two independent variables in the empirical equation. The reason for the lower accuracy at 100 cm at Nunn (Table 4) is likely caused by a biased estimation of the thermal conductivity for soils with high sand content and low soil moisture. According to the classical heat transfer equation, the main factor influencing the soil temperature is the thermal conductivity when the two boundary conditions (soil surface and soil bottom) are well estimated (Bittelli et al., 2015). This underestimation of soil temperature at deep depth could

lead to an underestimation of decomposition rate and over-prediction of SOC stock. However, at the NEON CPER site (close to Nunn site in distance and similar in soil texture), SOC at deep depths was slightly underestimated. This may again be related to the saturation limit of soils with high sand content discussed previously.

Our simulation accuracy of soil water is similar to other models (e.g. Shelia et al., 2018). It is common to see lower prediction accuracy for SWC in deep soil layers in modelling studies (e.g. Zhang et al., 2018). A model like MEMS 2.0 that receives only

daily precipitation input without rainfall intensity data cannot very accurately estimate the amount of infiltration and surface runoff, especially for large rainfall events. The SWC at deeper depths is impacted by this inaccuracy to a greater extent than the upper soil layers. For example, when infiltration was underestimated, the topsoil was filled with water regardless of the rainfall intensity, but the deep soil may not receive any water as the downward flow did not reach deeper depths. Additionally, for a 1D model, it is not possible to capture the slope and hill position effects on lateral flow of water. The use of pedotransfer





equations, which carry significant uncertainty in estimating soil hydraulic properties (Saxton and Rawls, 2006), added

uncertainty and contributed to the bias. The SWC at 100 cm at Rogers Farm site was constantly close to saturation (Fig. S3)

while the model predictions were at field capacity. It is likely that there was a relatively high water table at that site (no

measurements available) because otherwise the SWC at 100 cm would be at field capacity for a free drained soil. However,

for a 1D model, water table depth is needed to be provided as an input to simulate a high water table, and this information was

not available for the Rogers Farm site. As none of the NEON sites used in this study has a shallow water table, it is not possible

to test the model accuracy on SOM dynamics in such situations.

Eddy Covariance tower observations provide estimations of C and water exchange between the atmosphere and the ecosystem

(plant and soil). The measurements from the EC tower at the Konza Prairie used in this study showed lower GPP, ER, and ET

in 2009 compared with the other two years (Fig. 7). However, the mean air temperature and precipitation in 2009 and

specifically in the growing season of 2009 did not suggest any major differences from the other two years. Therefore it is not

possible to explain the pattern of measured GPP, ER, and ET across the years with the MEMS model, i.e., the simulated results

for these fluxes in 2009 were similar to those in the other two years. Specifically, the model slightly underestimated NEE in

the summer growing seasons of 2008 and 2010, which could be caused by an underestimation of autotrophic respiration from

plants. Regardless of this error, the simulated fluxes matched the observations and the accuracy was comparable to or better

than other published modelling studies (Chang et al., 2013; Schaefer et al., 2012; Yan et al., 2015).

Our sensitivity analysis showed the parameters of the maximum depolymerization rate of POM and maximum decay rate of

stable MAOM were among the most sensitive. The calibrated values for these two parameters were 0.0033 and 0.00034,

respectively, which indicated the average turn over time for MAOM was an order of magnitude slower than POM. This agrees

with a wealth of experimental data quantifying average turnover times of these two SOM fractions (von Lützow et al., 2007;

Tiessen and Stewart, 1983).

Parameters related to soil temperature and moisture effects on decomposition were also relatively sensitive. We used a

temperature effect curve (which served as a modifier in Equation 1) similar in shape to that in MEMS 1.0, but used a different

equation with fewer parameters (Equation S24 and Figure S4). Both curves assume the slope of the decomposition rate curve

(reflecting the sensitivity of decomposition to temperature) decrease at very high temperatures, which is different from a Q10

curve with a consistent increase in the slope of reaction rate with temperature. This decrease in the sensitivity of decomposition

rates to temperature at high temperature is supported by field observations (e.g., Del Grosso et al., 2005) and our calibrated

temperature effect curve was very similar to the recently calibrated curve used in the DayCent model (Gurung et al., 2020)

(Fig. S4). Our calibrated soil moisture effect curve was also similar to the curve used in DayCent (Fig. S4).

In a sensitivity analysis of the DayCent model, Gurung et al. (2020) found the top four most sensitive parameters for soil C

are the maximum decomposition rate of passive pool, the maximum decomposition rate of slow pool, the two parameters

controlling the curve of  temperature effect on decomposition. The passive and slow pools are the two largest soil C pools in

the DayCent model. This is similar to our results for the MEMS 2.0 model, where the most sensitive parameters were the decay

rates of the POM and MAOM pools and the temperature effect on decomposition. The role of temperature in controlling





decomposition is much higher than the other abiotic factors in both models. By studying several SOM models, Sierra et al.

(2015) found that temperature has a stronger control than moisture on the sensitivity of decomposition rates, which is also supported by experimental evidence (Reichstein et al., 2005; Yuste et al., 2007).

Following the same design concept of MEMS 1.0, we aimed to keep the model as parsimonious as possible while representing the new understanding of SOM dynamics and litter decomposition. Here we describe what we consider to be the main limitations of the model, which we will address during future model development and improvement.

In the current model version, large ungulate grazing is not explicitly modelled, but we plan to incorporate representation of grazing management in the future. In a global review of grassland, McSherry and Ritchie (2013) found that moderate grazing may increase or decrease total soil C depending on grass type and the mean effect was generally within ±10%. Even for heavy grazing, the mean effect is around ±15%. Grazing impacts ecosystem C and N dynamics in many ways, including by removing aboveground plant biomass and reducing plant litter (Piñeiro et al., 2010), modifying plant growth rates and biomass allocation

(Matches, 1992; Wei et al., 2011), changing root exudation (Hamilton et al., 2008; Sun et al., 2017), and adding C and N inputs from faeces and urine (Matches, 1992; McSherry and Ritchie, 2013). Management practices such as adaptive multi-paddock grazing, which may alter soil C and N stocks compared with continuous grazing (Byrnes et al., 2018), need to be accurately modelled to fully understand C dynamics in grassland.

We used the same temperature responses for all SOM fractions in MEMS 2.0. The differential temperature responses of SOM

fractions have been suggested (Conant et al., 2011; Davidson et al., 2000; Davidson and Janssens, 2006) and demonstrated in past studies (Benbi et al., 2014). Results are not consistent, however, especially between lab and field studies, and more data are needed to accurately quantify the differences and provide measurements for model calibration. Specifically, measurements of temperature effects on processes involved in POM and MAOM formation and decomposition, such as CUE, enzyme activities, and microbial respiration rates, would enable more accurate representation of temperature effects on soil C and N

dynamics in future versions of the MEMS model.

MEMS 2.0 includes two pools of MAOM, characterized as "exchangeable" and "stable", with the underlying assumption that a certain fraction of the MAOM pool is associated with the mineral surface with week boding and thus exchangeable (Kleber et al., 2007). A parameter (Frac_MAOMExchangeable) was used to define this fraction when both eMAOM and sMAOM were saturated. The value of this parameter was selected in our Bayesian optimization with a wide prior range because there

is currently little data on which to base this partitioning. When the pools are not saturated, this fraction is dynamic and it ranged between 14% and 27% in the top soil layers of the NEON sites in our simulations, with a trend an increase at deeper depths. Experimental data are needed to verify these findings. One hurdle to producing these data are reaching scientific consensus around a reliable method for measuring exchangeable MAOM. We plan to verify and re-parameterize the partitioning of MAOM between exchangeable and stable pools in the future as these data become available.

Currently, microbiota is modelled as a single entity in each model compartment, despite the significant differences between microbial groups in growth forms, life strategies, biomass stoichiometry, substrate preferences, and many traits that influence C and N cycling. We plan to increase the complexity of the microbial aspects of the model in future model versions, beginning





with separation of bacteria and fungi, which will include representing arbuscular and ectomycorrhizal systems separately because of their highly contrasting traits (Graaff et al., 2010; Hodge et al., 2001). In future model versions, bacterial and fungal

pools will have different N demands, CUEs, growth rates, substrate preferences, pH preferences, responses to disturbance, and necromass contributions to plant litter and SOM pools.

Soil pH is a static soil property in MEMS 2.0, but pH is dynamic in nature and has important effects on plant growth (Islam et al., 1980), microbial activity (Walse et al., 1998), and SOM dynamics (Averill and Waring, 2018). The grassland sites in our simulation have close to neutral pH, and it is thought to be relatively stable through time. However, representing other sites,

especially those where shifts in pH may be a major driver of C and N dynamics (e.g. fertilized systems), and will require dynamic modelling of pH and its effect on decomposition. This is a major goal of future model development efforts. Furthermore, soil mineralogy and redox conditions play important roles in mediating MAOM formation and persistence (Hall et al., 2015; Huang and Hall, 2017; Kögel-Knabner et al., 2008), but it is currently not included in MEMS 2.0.

Finally, MEMS 2.0 has been developed, calibrated, and tested using temperate grassland sites, and further development and

testing for different ecosystem types and climatic regions are needed to increase the utility of the model.

## 5. Conclusions

MEMS 2.0 is an ecosystem model built on the same foundational principles as MEMS 1.0: 1) the use of measurable pools that can be directly validated, 2) the MEMS hypothesis linking litter chemistry to microbial substrate use efficiency and SOM formation and 3) saturation behaviour of the MAOM pool. In developing MEMS 2.0, we created a full ecosystem model and

expanded upon the foundational principles to incorporate updated understanding of SOM dynamics such as the point-of entry framework (Sokol et al., 2019) and *in vivo* and *ex vivo* pathways of SOM formation (Liang et al., 2017b), through the entire soil profile. The resulting model represents ecosystem C and N pools in grassland systems across a wide range of climates and soils. In addition to calibrating and verifying the model in diverse systems beyond grasslands, future development will also aim to incorporate additional controls on SOM dynamics such as management impacts and additional environmental factors

including variable pH and redox conditions, mineralogy, soil microbial community structure, and temperature sensitivities. Much of the planned development hinges upon the availability of data quantifying these relationships across ecosystems, soil types, and soil depths, and we encourage experimentalists to continue this important work and disseminate their results to bolster modelling efforts such as this one. Using mechanistic models with measurable pools allows for detailed hypothesis testing to improve understanding of SOM dynamics, and will ultimately provide more reliable predictions of land-climate

feedbacks for use in land management and global change mitigation.

**Code and Data availability**

The NEON data set is available at https://data.neonscience.org/. The SCAN data set is available at https://www.wcc.nrcs.usda.gov/scan/. The EC data used in this study can be found on the website of the University of Kansas (http://www.dept.ku.edu/~biomet/KU_Biometeorology_Lab/Data.html). The MODIS data product is hosted at the Oak Ridge

National Laboratory (https://modis.ornl.gov/). The SOM fraction data of the six NEON sites and model input files used in this study can be accessed at the Zenodo online repository (DOI 10.5281/zenodo.4404685). The source code of the MEMS 2.0 model will be made available on GitHub after the completion of the supporting documents (e.g., model user manual).

**Author contribution**

MFC, YZ, JML, ADR, KP, and SO contributed to the conceptualization of the model structure and wrote the manuscripts. YZ
formulized the equations and coded the model. YZ and JML collected the data used in the simulations. RE conducted the soil organic matter fractionation work. MFC, JML and RE interpreted the lab data. MFC, KP, and SO contributed substantial interpretation and discussion, and supervised the study.

**Competing interests**

The authors declare that they have no conflict of interest.

**Acknowledgements**

The project was supported with funding from the US DOE Advanced Research Projects Agency-Energy programme (ROOTS project; DE-FOA-00001565), from the NSF DEB (award n.1743237) and from Shell Inc. (contract n.4550183252). The authors
like to thank Dr. William J. Parton for model development discussions. The authors also thank Dr. Peter J. Ross for providing his Fortran source code of the soil water and solute transport model.

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
