# Peer review of "Simulating measurable ecosystem carbon and nitrogen dynamics with the mechanistically-defined MEMS 2.0 model"

_Biogeosciences, 2020_

## Author Comment (AC1)

[Figure]

Figure 1: Preliminary results of the verification of MEMS 2.0 and DayCent (a version previously calibrated for grassland) soil organic C simulations, using the data measured in this study

---

## Author Comment (AC2)

[Figure]

Figure 1: Preliminary results of the verification of MEMS 2.0 and DayCent (a version previously calibrated for grassland) soil organic C simulations, using the data measured in this study

---

## Author Response (AR1)

**Reviewer 1.**

The authors present a new full ecosystem model MEMS2.0 by extending a previous soil biogeochemical model MEMS1.0, which was constructed based on measurable carbon pools. This extension includes model components of aboveground and rhizosphere processes, vertical transport /mixing in the soil column and N cycling. The model has the capability of including litter chemistry, dynamic CUE and CN ratios, as well as MAOM saturations. They first validated a subcomponent, the litter decomposition model using year-long laboratory incubation experiments and further validated the full model with long-term field observation data. Adding the fire frequency impact is interesting, but it remains unclear how it was done.

Overall, I appreciate the effort of extending a novel modeling concept using measurable pools. I think this manuscript would significantly benefit from more work that better structure its discussion, clarify its message with an emphasis on "novel" model capabilities it's providing.

*Response: We greatly appreciated the reviewer's comments. We reconstructed the discussion and added sub-headings to make the message clearer as the reviewer suggested. The fire impact is included because we represent grasslands which are impacted by fire. However, a full and detailed representation of fire impacts is beyond the scope of this study. Thus, in this version we adapted the fire representation from the DayCent model. During a fire event, plant biomass and litter are removed and a fraction of the biomass/litter C and N is returned to the soil as unhydrolysable C and mineral N. Since fire frequency determines the amount of C and N loss in fire, it impacts the amount of C and N input to soil. Our experience with DayCent modeling showed significant impact of fire frequency on grassland soil C and N. We acknowledge in the text that this is a simplistic representation of fire impacts, we provided more details to clarify how we represented it in L352-354 (track-change version of the revised manuscript).*

Some major technical notes:

1. The model description section needs further clarification (section 2.1).

It is not clear to me how the continuous soil horizons are divided into layers (line 124-126), and whether the rhizosphere and bulk soil separation has been done for all horizons/layers. How deep the conceptual rhizosphere goes? Since the authors mentioned the differences between subsoils (> 30 cm deep) and topsoils, it would be useful to make clear separations in the model description.

*Response: The user-defined horizons are multiples of 5 cm for the top 50 cm, and multiples of 10cm for below 50 cm. The model has fixed depths for layers: 0 - 2cm, 2 - 5cm, then 5 cm increments for 5 – 50 cm, and 10 cm increments for layers below 50 cm. For example, if a horizon is 20-35 cm, it will be divided into three layers with 5 cm per layer. The conceptual rhizosphere goes as deep as the root system so it is in every soil layer where roots go as deep. In the model, there is no separation between the topsoil and subsoil. We add details to more clearly*

*describe the soil horizons and rhizosphere structure in L133-137 and added rhizpshere depth in L138.*

The CN coupling part needs some clarification as well. Inorganic N (nitrate or ammonium or both?) will be assimilated by microbes if the substrate N could not meet microbial N demand. Is this modeled implicitly? I have looked through table S2 and S3 and wondering if that's represented through MicCNeff.

*Response: Both nitrate and ammonium are assimilated by microbes and there is no preferential uptake. The uptake of inorganic N is modeled explicitly. The model first estimates a CUE based on the availability of N in the organic substrate and in the inorganic N pool in the soil (Equation S18). Then it calculates the potential demand for inorganic N based on the amount of carbon assimilated and the minimum C/N ratio of microbes. The actual uptake is a result of competition with the plants (hourly time step calculation based on their demand). We describe this in the text (Lines 156 - 160), and we revised the text to be clearer in L172-176 and added the equation to estimate the potential demand of inorganic N to Table S2.*

2. Description of observational datasets could be more balanced and condensed.

The leaf litter incubation experiments were used to validate the litter decomposition model, but there's no specific description of the CN ratios in the method or result section. Specifically Figure 3 and line 350, what are the CN ratios or the range of CN ratios in the experimental data?

*Response: The CN ratio data were published in Soong et al. (2015). The CN ratios are 10.8, 36.1, 52.8, 92.3, and 126.6 for alfalfa, oak, ash, bluestem, and pine, respectively. We added these data to the Method section in L277-278.*

The description of soil OM fractionation for modeling could be condensed (line 257-281). Currently the mix of data description (DOM, POM and MAOM) and detailed experimental approaches make it difficult to find key information. On a side note, I also find it is inconsistent with OM pool descriptions: here the author mentioned MEMS2.0 represents three bulk soil OM fractions (line 262), but in the model description there were five (line 181), and in Figure 1 bulk soil box, do you consider three or four OM pools? Additionally, it would be useful to explain how eMAOM and sMAOM are determined/parameterized.

*Response: We are sorry for this confusion. We condensed the description of soil OM fractionation, to make it clearer that the model includes five OM pools which comprise the bulk soil (see L283-320). However, our lab measurement did not separate sMAOM and eMAOM. As we discuss in the text, "one hurdle to producing these data is reaching scientific consensus around a reliable method for measuring exchangeable MAOM. We plan to verify and re-parameterize the partitioning of MAOM between exchangeable and stable pools in the future as these data become available." (Line 577 – 579 in the Discussion of first submission). The*

*parameterization of the fraction of eMAOM and sMAOM is achieved through the Bayesian optimization, which we describe in the Discussion (Line 573 – 575): "A parameter (Frac_MAOMExchangeable) was used to define this fraction when both eMAOM and sMAOM were saturated. The value of this parameter was selected in our Bayesian optimization with a wide prior range because there is currently little data on which to base this partitioning." We now realize that we should have presented this earlier, and in more detail. We revised text to explain this in the Method section for clarity (L318-320), and revised the Discussion section (L620-631).*

3. The discussion section would benefit from more structured statements

Currently the section is a mixed discussion of simulation results, model performance, and model formulation. There's no clear trail describing the key features of this new model and how well each new component behaved. Alternatively, I suggest using a few subtitles to describe the key model features, within each subsection, maybe add a short summary of the model formulation, followed by model performance and sensitivity discussion, then model limitations.

Additionally, I do not find the model performance comparison very compiling (e.g. paragraph starting from line 479). Just like what the authors mentioned in the former paragraph, model input data have a huge impact on the final predictions. Vegetation, NPP, climate, soil texture, historical temperature and moisture, there are many factors that contribute to model uncertainties. Since all other models mentioned in the paragraph were validated using different data, there's no basis for accuracy comparison. Alternatively, maybe add an additional simulation with a selected dataset that one of these models used, so the climate and vegetation data driving SOM dynamics are associated with the same level of uncertainties.

*Response: We appreciate the reviewer's suggestions to improve the structure of the Discussion. We reconstructed the discussion section as the reviewer suggested, adding sub-headings, and focusing on the new model key features. We agree that a robust model comparison requires that all models simulate the same sites. However, we feel a formal model intercomparison is beyond the scope of this initial model description and demonstration paper, and we are planning to perform these comparisons in subsequent analyses/publications. Since reviewer two is also interested in a model intercomparison, we run and present here a simple preliminary comparison with DayCent (see figure below). We used a version of DayCent previously calibrated with grassland sites. Since DayCent only represents the top 20cm soil depth, the SOC observations of the six NEON grassland sites and MEMS 2.0 results were summarized to the top 20 cm. These preliminary results show that the MEMS model performs well on all six sites, while the DayCent model significantly under-estimates SOC on the OAES site. However, since DayCent was not calibrated with the same data set, we do not consider these data robust enough to be included in this paper, and will conduct a proper comparison at a later stage. Because DayCent only simulates the total SOC in the top 20 cm soil, more sites are needed to provide a good number of observations for model calibration and validation. We deleted some of the model comparison discussion in the Discussion section to avoid confusion.*

[Figure]

*Figure 1: Preliminary results of the verification of MEMS 2.0 and DayCent (a version previously calibrated for grassland) soil organic C simulations, using the data measured in this study*

minor note: There's no space after each paragraph or each reference, which significantly impacted the readability of this manuscript. All equations were summarized in a table that is tightly fitted with a small front size, consider changing the format to increase readability. Maybe add proper subtitles, like MAOM dynamics, Microbial dynamics, environmental variable etc.

*Response: We are sorry about that. We edited the text for improved readability by adding space between paragraphs and changing the format.*

I would also suggest releasing the model code for review.

*Response: Currently, we are filing for an invention disclosure, and after that we'll make the code available.*

Some minor comments:

line 40: The model comparison is a bit misleading. Many models included are microbial explicit models that are designed to capture the biological control on SOM dynamics. The modeling goals are different, and we do not necessarily need all the models to address all of the needs (line 48). In particular, MEMS 2.0 is not a microbial explicit model.

*Response: We agree that MEMS 2.0 is different from the typical microbial-explicit models. It uses first-order kinetics but the decomposition rate is affected by the C/N ratio of the microbial pool. MEMS 2.0 also uses dynamic CUE, which makes it different from the traditional SOC models. These features (CUE and C/N specifically) of the discrete microbial pools in MEMS 2.0 provide feedbacks to the fluxes and decomposition between and from other model pools. Although different from existing microbially-explicit models, we feel these feedbacks make MEMS 2.0 microbially-explicit in some ways. Additionally, as described in the paper, we*

*developed and tested a stand-alone litter decomposition model that used microbial biomass as a decomposition rate modifier (as used in the CORPSE model). However, statistically that version did not provide better overall fit to the Soong et al. 2015 data than the current structure in MEMS 2.0. This all said, we take your point that we may not need to point out a comparison of distinctly different model structures because as you rightly state, we do not need all models to address all needs. We added a section in Discussion to address the mocrobially-explict and first-order models in L559-575.*

line 59: SOM pool partitioning is only one of the reasons for large model disagreement, see Sulman et al. https://link.springer.com/article/10.1007/s10533-018-0509-z

***Response****: We agree – there are many reasons why bulk SOM measurements may disagree with model predictions. It was not our intention to say that partitioning was the only, or even major driver, but can be an important determinant in understanding where a model is misrepresenting the mechanisms/fluxes. We have modified this sentence to emphasize this point, including adding the Sulman et al reference in L64-65.*

line 75: There're a few other CN coupled microbial explicit models besides MIMICS-CN, such as MEND-CN (https://www.sciencedirect.com/science/article/pii/S0022169420302377), ORCHIMIC (https://gmd.copernicus.org/articles/11/2111/2018/), BAMS2 (https://link.springer.com/article/10.1007/s10533-019-00580-7)

***Response****: We are sorry we missed them. We added these models to our Table S1.*

line 96-100: Sokol et al's GCB concept paper was mentioned several times through the manuscript, it would be good to discuss if this model can be used to validate this conceptual model at the ecosystem scale

***Response****: This is a very good suggestion. However, it is hard to address this without conducting an ecosystem scale modeling study. We could add this in our future work.*

Line 165: consider add N flow in figure 1

***Response****: We are not sure what the referee is asking here. The N flow is in the figure and we feel extra arrows to call out C and N separately would make the figure too 'busy'. All the organic matter pools have N in them. Mineral N leaching, uptake by plant, immobilization and mineralization are in the figure.*

Line 308: consider presenting the fitted saturation function, because it was discussed as a key point --MAOM saturation

*Response: We added a figure in the Supplementary Material to show the fitted function (Fig. S2).*

Line 360: consider adding a second y-axis to show the two data series more clearly

*Response: The suggestion of adding a second y-axis is good. However, we feel it could make the figure busy and affect the readability and would prefer to keep the figure as it is.*

Line 390: explain " diagnostic soil horizon"

*Response: A "diagnostic" horizon is a horizon used to define a soil taxonomic unit. However, we see now that it is jargon, and we deleted "diagnostic".*

Line 456: need some justification, like adding MAOM saturation limit

*Response: We will added the saturation limit values in L538-539.*

Line 484: "serval" should be "several"

*Response: It is corrected.*

Line 605: dynamic pH and redox, microbial community structure etc sounds a little far away in terms of current model structure and focus

*Response: We intend to point to the limitation of the current model, as well as its potentials. We consider these limitations of the current version, and plan to address them in future developments. We actually already have projects in the pipeline to do this work, so we do not see it as far away as the referee thinks. Therefore, we would like to keep this section. However, we can certainly omit this part of the discussion if preferred.*

**Reviewer 2**

This manuscript describes the MEMS 2.0 model, which combines a soil organic matter module based on measurable pools of POM and MAOM with vegetation, soil heat, and soil water components. The model is calibrated and validated using data from grassland sites, with pools of POM and MAOM along with eddy covariance measurements of $CO_2$ fluxes as the metric for evaluating the model. Overall, the paper is well written. The Introduction especially is an excellent review of current progress and limitations of soil organic matter models and makes a good case for the reasoning behind developing the MEMS 2.0 model.

In my opinion, the weakness of the manuscript is that the only results presented are the evaluation of the model itself. The manuscript makes a case for developing a model with measurable soil pools, but does not provide any new insights about the results of the model or whether the model provides any benefit for ecosystem prediction or process understanding compared to other current ecosystem models. The paper is a well written model description, but is very limited in terms of new scientific insights or new results.

*Response: We appreciate the reviewer's opinion but do feel that the scientific insights and new results provided in the paper are important and worthy of publication. MEMS 2.0 is different from other models in its design of flows and controls which is critical for a SOM model. We incorporated several recent understandings of the decomposition and protection processes (such as the MEMS theory, point of entry, saturation, in vivo/ex vivo pathway), which make the model unique. Additionally, MEMS 2.0 is a full ecosystem model with SOM simulated to deeper depths while some of the other new generation models only simulate one layer of soil and do not have a full plant module or water module (more practical to be used as a tool for scientific inquiry and real world decision making). In this study, we also present new and original measured and modeled SOM fraction data from several sites with deep depths.*

*We also think a model description paper without a comparison to other models (using the same data to calibrate and validate different models) is highly valuable for publication. Numerous model description papers have been published and many are highly cited* (e.g., Davidson et al., 2012; Parton et al., 1998; Robertson et al., 2019; Wang et al., 2013). *Please see our response to the other reviewer regarding our plan of a full proper model comparison and a preliminary comparison with the DayCent model (Figure 1). The response to reviewer one is copied here. "We used a version of DayCent previously calibrated with grassland sites. Since DayCent only represents the top 20cm soil depth, the SOC observations of the six NEON grassland sites and MEMS 2.0 results were summarized to the top 20 cm. These preliminary results show that the MEMS model performs well on all six sites, while the DayCent model significantly under-estimates SOC on the OAES site. However, since DayCent was not calibrated with the same data set, we do not consider these data robust enough to be included in this paper, and will conduct a proper comparison at a later stage. Because DayCent only simulates the total SOC in the top 20 cm soil, more sites are needed to provide a good number of observations for model calibration and validation."*

*We reconstructed the discussion to make the new scientific insights and results more clearly presented and emphasized.*

[Figure]

*Figure 2: Preliminary results of the verification of MEMS 2.0 and DayCent (a version previously calibrated for grassland) soil organic C simulations, using the data measured in this study*

In terms of the structure of the model, MEMS 2.0 is compared to other existing models including both first-order models like CENTURY and microbial-explicit models like MIMICS, CORPSE, and MEND. One key point that I thought was overlooked was the role of microbial biomass in MEMS 2.0. The model does include a microbial biomass pool, but all decomposition reactions are first-order. Thus, in contrast to microbial-explicit decomposition models, MEMS 2.0 does not include any interaction between microbial biomass and depolymerization or decomposition rate. I would suggest acknowledging this explicitly when comparing MEMS 2.0 to microbial-explicit decomposition models to make it clear to readers.

**Response:** *The referee is right in that the microbial biomass does not have a direct effect on decomposition rate in MEMS 2.0. However, different from CENTURY model, the C/N ratio of the microbial pool in MEMS 2.0 is a modifier to the decomposition rate (the $MicCN_{eff}$ in Table S2 equations). We made a version of the stand-alone litter decomposition model that used microbial biomass as a decomposition rate modifier, as in the CORPSE model. However, statistically that version did not provide better fit to the Soong et al. 2015 data than the current structure in MEMS 2.0. The method of explicitly using microbial biomass has its advantage of quick response to carbon input (priming effect) compared with the first order method, but the first order equation is more stable in longer-term simulations. Luo et al., (2016) has stated that the microbe-explicit models have "oscillatory responses to perturbation and insensitivity of soil C storage to C input, in comparison with classic models"; "such patterns may exist at the microbial reaction sites but have not been observed in litter decomposition and soil incubation studies.". The reasons are that first, the microbial biomass is very hard to predict (most models only use a constant death rate); second, there is a community of microbes that are functioning very differently (treating them as one uniform species is not correct); third, it is the amount of active enzyme rather than microbial biomass that is correlated with decomposition rate (using microbial biomass to approximate enzyme amount is not accurate) (e.g., Li et al., 2018). A sound*

*microbial-explicit model should model both the microbial community and enzyme activity but it will make the model very complex and hard for large scale applications due to the limitation of data availability. After weighing the different factors, we decided to use the stable first-order approach to make a parsimonious model structure while introducing new concepts (microbial C/N control on decomposition and dynamic CUE) to improve it from the traditional first-order models. We added a section in the Disucussion in L560-574 (track-change version of the revision) to make it clear to the readers that MEMS 2.0 is different from the microbial-explicit model like CORPSE and also different from CENTURY.*

Given that MEMS 2.0 uses a first-order decomposition framework similar to other models like CENTURY, it's unclear what the major conceptual advance of the soil model is, beyond being able to compare certain model pools more direclty with measurements. Is the model significantly different from a reconfiguration or reparameterization of a CENTURY-like model using POM and MAOM as calibration targets? Would this model produce different projections of SOM dynamics than CENTURY would if they were both calibrated to a similar dataset? The only information that the manuscript really provides is a demonstration that MEMS 2.0 can be calibrated successfully, along with an assertion that the measurable MAOM and POM pools represent an improvement on previous models. But apart from the a priori statement that it is better to have measurable pools, can the actual added value of the MEMS 2.0 model be demonstrated? Some more discussion or demonstration of how the MEMS 2.0 model is expected to advance the biogeochemical modeling field in concrete terms of actual model results or predictions (beyond the relatively abstract concept that measurable pools are inherently better) would help to provide better context for how the model developments represent a scientific advance rather than a purely technical advance.

*Response: We agree with the reviewer that a CENTURY-like model can be calibrated using measurable fractions of SOM. In fact, at least two studies have done that (Luo et al., 2014; Skjemstad et al., 2004). However, a SOM model is not simply about the decomposition equation (no matter first-order or microbial explicit). The design of the flows and the controls are critical. In MEMS 2.0, we aim to model the real processes (as best as we can/understand them) instead of merely predicting variables with a black box. For example, the model represents two pathways of SOM formation, maintaining separate flows for soluble and structural inputs which is very different than the CENTURY model. Representing new understanding and theories helps the advance in biogeochemical modelling. In our opinion, MEMS 2.0 advances biogeochemical modelling also by allowing us to test new hypotheses. For example, MEMS 2.0 can simulate the impacts of litter quality on rate of MAOM saturation in a subsoil horizon with distinctly different soil texture than topsoil. We modified the Discussion section to emphasize the features of our model.*

In terms of the model description: Figure 1 shows the rhizosphere and bulk soil as separate model compartments. However, there is only a full description of bulk soil dynamics (Section

2.1.2). The dynamics of the rhizosphere also need to be described. If the description of the rhizosphere in the litter decomposition section is the full explanation, then perhaps that section should be renamed "Litter layer and rhizosphere" to avoid confusion.

*Response: The reviewer is correct; the description of the rhizosphere is included in the litter decomposition section. We modified the section title to avoid confusion as the reviewer suggested.*

Specific comments:

Line 126: Soil horizons are divided into "thinner layers" but the actual layer thickness is not provided

*Response: The other reviewer raised a similar question. As reported above, the user-defined horizons need to be multiples of 5 cm for the top 50 cm, and multiples of 10cm if lower than for below 50 cm. The model has fixed depths for layers: 0 - 2cm, 2 - 5cm, then 5 cm increments for 5 – 50 cm, and 10 cm increments for layers below 50 cm. For example, if a horizon is 20-35 cm, it will be divided into three layers with 5 cm per layer. We modified the text to more clearly describe the soil horizons and rhizosphere structure in L133-138.*

Line 159: The description should also state here what happens when there is insufficient mineral N for immobilization. In that case, is CUE reduced? Or does decomposition slow down?

*Response: When there is insufficient mineral N for immobilization, both CUE and C/N of the microbial biomass reduces. The reduced microbial C/N leads to a reduced decomposition rate. We added this to L177-178.*

Table 2: The notation in this table is different from the notation used in Equations 1 and 2 and the equations in Table S2, which have the same parameters but use different variable names. Using two different names for these parameters makes it difficult to tell which parameter in the actual equation is being referred to.

*Response: We thank the reviewer for noticing this issue, and are sorry we had not noticed it before. The parameter names in Table 2 were the names used in the input files thus different from the notation in the equations. We added a new column in Table 2 with corresponding symbols used in the equations. The readers can look up the parameters used in the input file and symbols used in the equations.*

Line 467: Could the underestimation of POM C in deeper layers be related to rooting depth? If actual rooting depth is deeper than assumed in the model, the model would be underestimating fresh POM production at that depth

*Response: We agree it is possible, and we added this interpretation. Rooting depth is very uncertain according to the measurements. Studies have reported very different values at the same sites (methods were different). We used our best estimation based on NEON root biomass measurements and other studies conducted in the same areas. Added this explanation of root depth and distribution in Line 552-554*

Line 470-471: There are multiple previous studies demonstrating that POM decomposition is slower in deep soils, and arguing that this could be due to limited microbial activity: e.g., Hicks Pries et al. 2018, Fontaine et al. 2007. These previous papers have suggested that the lower decomposition rate of POM at depth could be due to lack of fresh organic matter inputs to prime microbial decomposition. This is an area where microbial-explicit decomposition models have been invoked to explain the variation in decomposition rates via priming effects (e.g., Hicks Pries et al., 2018).

*Response: We agree that the typical microbial-explicit models have the advantage of reproducing the trend of decomposition in some priming effect experiments. However, it has weaknesses as described above and in Luo et al. (2016). Studies showed the quality of substrate and N availability can drive priming effect; however, the mechanism of priming effect are still not clear  (Li et al., 2018), and studies found no robust correlation between soil microbial biomass and priming (Liu et al., 2017). We will keep following advancement in the understating of the priming effect, and will modify the model to represent it, when the scientific community has reached a better mechanistic understanding of it. We added a section in the Discussion to address this (L560-574).*

Line 485-486: Sulman et al. (2014) used MAOM measurements from the Duke and Oak Ridge National Laboratory elevated CO2 experiments in model validation. So, this is one example of a model that used soil fraction measurements from more than one site. I do agree that this manuscript uses more fraction measurements from a greater number of sites than previous studies.

*Response: Sorry for this oversight. We appreciate this comment, and we changed Table S1 for CORPSE model (verified with two sites). The Sulman et al. (2014) reference is added in L508-509.*

Line 495: The FUN-CORPSE model did use both C and N measurements for validation, including N mineralization rates and soil %N.

*Response: We changed "they" to "most of them" in Line 607. Table S1 already has FUN-CORPSE.*

Line 532: The units should be provided for the two parameter values

*Response: We added units in Line 517*

Line 540-541: The reduction in sensitivity of decomposition to temperature at high temperature does seem to be justified. However, since the soil temperatures in these simulations did not exceed 20-25C (Fig. S2), it does not seem like the model calibration would have provided much constraint of that high temperature area of the response curve. The response only really flattens at temperatures over 25-30 C.

*Response: We agree that the sites used in this study do not have soil temperature above 25 C. However, the summer soil temperature exceeded 20 C. The temperature curve started to flatten right after 20 C. We will simulate tropical and subtropical sites to better characterize the curve at high temperature in future studies.*

Line 572: "week boding" - should this be "weak binding"? Or "weak bonding"?

*Response: Thank for catching this. It should be "weak binding".* Changed in Line 621

References from reviewers:

Fontaine, S., Barot, S., Barre, P., Bdioui, N., Mary, B., & Rumpel, C. (2007). Stability of organic carbon in deep soil layers controlled by fresh carbon supply. Nature, 450(7167), 277-U10. http://www.nature.com/nature/journal/v450/n7167/abs/nature06275.html

Hicks Pries, C. E., Sulman, B. N., West, C., O'Neill, C., Poppleton, E., Porras, R. C., et al. (2018). Root litter decomposition slows with soil depth. Soil Biology and Biochemistry, 125, 103–114. https://doi.org/10.1016/j.soilbio.2018.07.002

References from authors:

Davidson, E. A., Samanta, S., Caramori, S. S. and Savage, K.: The Dual Arrhenius and Michaelis–Menten kinetics model for decomposition of soil organic matter at hourly to seasonal time scales, Global Change Biology, 18(1), 371–384, https://doi.org/10.1111/j.1365-2486.2011.02546.x, 2012.

Li, L.-J., Zhu-Barker, X., Ye, R., Doane, T. A. and Horwath, W. R.: Soil microbial biomass size and soil carbon influence the priming effect from carbon inputs depending on nitrogen

availability, Soil Biology and Biochemistry, 119, 41–49, https://doi.org/10.1016/j.soilbio.2018.01.003, 2018.

Liu, X.-J. A., Sun, J., Mau, R. L., Finley, B. K., Compson, Z. G., van Gestel, N., Brown, J. R., Schwartz, E., Dijkstra, P. and Hungate, B. A.: Labile carbon input determines the direction and magnitude of the priming effect, Applied Soil Ecology, 109, 7–13, https://doi.org/10.1016/j.apsoil.2016.10.002, 2017.

Luo, Y., Ahlström, A., Allison, S. D., Batjes, N. H., Brovkin, V., Carvalhais, N., Chappell, A., Ciais, P., Davidson, E. A., Finzi, A., Georgiou, K., Guenet, B., Hararuk, O., Harden, J. W., He, Y., Hopkins, F., Jiang, L., Koven, C., Jackson, R. B., Jones, C. D., Lara, M. J., Liang, J., McGuire, A. D., Parton, W., Peng, C., Randerson, J. T., Salazar, A., Sierra, C. A., Smith, M. J., Tian, H., Todd-Brown, K. E. O., Torn, M., Groenigen, K. J. van, Wang, Y. P., West, T. O., Wei, Y., Wieder, W. R., Xia, J., Xu, X., Xu, X. and Zhou, T.: Toward more realistic projections of soil carbon dynamics by Earth system models, Global Biogeochemical Cycles, 30(1), 40–56, https://doi.org/10.1002/2015GB005239, 2016.

Luo, Z., Wang, E., Fillery, I. R. P., Macdonald, L. M., Huth, N. and Baldock, J.: Modelling soil carbon and nitrogen dynamics using measurable and conceptual soil organic matter pools in APSIM, Agriculture, Ecosystems & Environment, 186, 94–104, https://doi.org/10.1016/j.agee.2014.01.019, 2014.

Parton, W. J., Hartman, M., Ojima, D. and Schimel, D.: DAYCENT and its land surface submodel: description and testing, Global and Planetary Change, 19(1), 35–48, https://doi.org/10.1016/S0921-8181(98)00040-X, 1998.

Robertson, A. D., Paustian, K., Ogle, S., Wallenstein, M. D., Lugato, E. and Cotrufo, M. F.: Unifying soil organic matter formation and persistence frameworks: the MEMS model, Biogeosciences, 16(6), 1225–1248, https://doi.org/10.5194/bg-16-1225-2019, 2019.

Skjemstad, J. O., Spouncer, L. R., Cowie, B. and Swift, R. S.: Calibration of the Rothamsted organic carbon turnover model (RothC ver. 26.3), using measurable soil organic carbon pools, Soil Res., 42(1), 79–88, https://doi.org/10.1071/sr03013, 2004.

Wang, G., Post, W. M. and Mayes, M. A.: Development of microbial-enzyme-mediated decomposition model parameters through steady-state and dynamic analyses, Ecological Applications, 23(1), 255–272, https://doi.org/10.1890/12-0681.1, 2013.

---

## Author Response (AR2)

From associate editor

There remain a number of minor issues to deal with (see reviews), and from my point of view one major one: the source for the model really needs to be made available at this stage; it's simply unacceptable to promise it but not provide a link in the paper. Doing so will increase the future use of this model, and improve scientific transparency and reproducibility.

**Response**: *We have addressed the few remaining minor issues raised by the referees and our punctual response is reported below. We agree with the editor regarding the benefits and importance of sharing source code. We plan to have the model open-source for research use as soon as possible. However, since we received requests for using this model by private companies, we are in the process of licensing the code. In the meantime, we are willing to share the code with peer scientists upon individual request. We thus changed the statement in the Code Availability section, which now reads: "The source code of the MEMS 2.0 model is available upon request".*

From Anonymous Referee #2

A few minor comments:

Line 233: I believe this should be Richards Equation, not Richard Equation

**Response**: *Changed to "Richards" (Line 233)*

Line 515: receives is misspelled

**Response**: *Changed to "receives". (Line 515)*

Line 548-549: While oscillations and insensitivity to inputs have been identified in the past as issues with some microbial-explicit soil carbon models, later versions of most models have largely corrected those issues, so I'm not sure it's accurate to describe this as still being a major issue in current microbial-explicit models. See: Georgiou, K., Abramoff, R. Z., Harte, J., Riley, W. J., & Torn, M. S. (2017). Microbial community-level regulation explains soil carbon responses to long-term litter manipulations. Nature Communications, 8(1), 1–10. https://doi.org/10.1038/s41467-017-01116-z

**Response**: *We thank the referee for this very important and relevant reference and are sorry we previously missed it. We added a new sentence in Line 549-550.* "Georgiou et al. (2017) proposed a way to reduce the oscillation and correct the insensitivity to C input by introducing a density-dependent formulation of microbial turnover."

Line 579: CORPSE is misspelled

**Response**: *Changed to "CORPSE". (Line 579)*